# A minimum curvature algorithm for tomographic reconstruction of atmospheric chemicals based on optical remote sensing

Sheng Li, Ke Du

Department of Mechanical and Manufacturing Engineering, University of Calgary, Calgary, AB T2N 1N4, Canada

*Correspondence to*: Ke Du (kddu@ucalgary.ca)

**Abstract.** Optical remote sensing (ORS) combined with the computerized tomography (CT) technique is a powerful tool to retrieve two-dimensional concentration map over an area under investigation. Whereas medical CT usually uses beam number of hundreds of thousands, ORS-CT usually uses a beam number of dozens, thus severely limiting the spatial resolution and the quality of the reconstructed map. The smoothness *a priori* information is therefore crucial for ORS-CT. Algorithms that

produce smooth reconstructions include smooth basis function minimization, grid translation and multiple grid (GT-MG), and low third derivative (LTD), among which the LTD algorithm is promising because of fast speed. However, its theoretical basis must be clarified to better understand the characteristics of its smoothness constraints. Moreover, the computational efficiency and reconstruction quality need to be improved for practical applications. This paper first treated the LTD algorithm as a special case of Tikhonov regularization that uses the approximation of the third-order derivative as the regularization term.

Then, to seek more flexible smoothness constraints, we successfully incorporated the smoothness seminorm used in variational interpolation theory into the reconstruction problem. Thus, the smoothing effects can be well understood according to the close relationship between the variational approach and the spline functions. Furthermore, other algorithms can be formulated by using different seminorms. On the basis of this idea, we propose a new minimum curvature (MC) algorithm by using a seminorm approximating the sum of the squares of the curvature, which reduces the number of linear equations to half that in

the LTD algorithm. The MC algorithm was compared with the non-negative least square (NNLS), GT-MG, and LTD algorithms by using multiple test maps. The MC algorithm, compared with the LTD algorithm, shows similar performance as in terms of reconstruction quality but requires only approximately 65% the computation time. It is also simpler to implement than the GT-MG algorithm because it directly uses high-resolution grids during the reconstruction process. Compared with the traditional NNLS algorithm, it shows better performance in three aspects: (1) the nearness of reconstructed maps is improved

by more than 50%; (2) the peak location accuracy is improved by 1- 2 m; and (3) the exposure error is improved by 2 to 5 times. Testing results indicated the effectiveness of the new algorithm according to the variational approach. More specific algorithms could be similarly further formulated and evaluated. This study promotes the practical application of ORS-CT mapping of atmospheric chemicals.

## 1 Introduction

Measuring the concentration distribution of atmospheric chemicals over large areas is required in many environmental applications, such as locating hotspots or emission sources of air pollutants (Wu et al., 1999), understanding air pollutant dispersion and airflow patterns, and quantifying emission rates or ventilation efficiency (Samanta and Todd, 2000; Belotti et al., 2003; Arghand et al., 2015). The traditional network method uses multiple point samplers placed at various locations in the region under investigation. This method is intrusive, time-consuming, and limited in temporal and spatial resolution (Cehlin, 2019). The advanced method is based on the combination of optical remote sensing and computerized tomography techniques (ORS-CT). ORS-CT is a powerful technique for sensitive mapping of air contaminants throughout kilometer-size areas in real time (Du et al., 2011). Two commonly used ORS techniques use an open-path tunable diode laser (TDL) and open-path Fourier transform infrared spectrometer. The ORS analyzer emits a light beam targeted at multiple mirrors, which reflect the beam back to the analyzer. For each beam path, the path-integrated concentration (PIC) is obtained. After multiple PICs are collected, a two-dimensional concentration map can be generated through tomographic reconstruction algorithms (Hashmonay et al., 2001). The ORS-CT method provides better spatial and temporal resolution than the network approach, and it is more sensitive than the range-resolved optical techniques. It is also non-intrusive and suitable for continuous long-term monitoring.

In ORS-CT mapping of atmospheric chemicals, owing to factors including system cost, response time, beam configuration, the number of beams is only tens, whereas the number of beams in medical CT is hundreds of thousands. The very small beam number poses several challenges in tomographic reconstruction algorithms. In practice, transform methods based on the theory of Radon transformation using a filtered back projection formula are not feasible because of noise and artifacts in the reconstructions (Radon, 1986; Herman, 2009). Series expansion methods, which discretize the reconstruction problem before any mathematical analysis, are usually used in ORS-CT. The underlying distribution is represented by a linear combination of a finite set of basis functions (Censor, 1983). The simplest type is the pixel-based approach, which divides an area into multiple grid pixels and assigns a unit value inside each pixel. The path integral is approximated by the summation of the product of the pixel value and the length of the path in that pixel. A system of linear equations can be set up for multiple beams. The inverse problem involves finding the optimal set of pixel concentrations according to criteria including the least square criterion to minimize the summation of the squared errors between the observed and model-predicted PICs; the maximum likelihood (ML) criterion to maximize the probability of the PIC observations given the distribution of the random variables of the concentrations and observation errors; and the maximum entropy criterion to maximize the entropy of the reconstructed maps, given that the average concentration of the map is known (Herman, 2009). Commonly used pixel-based algorithms are algebraic reconstruction techniques (ART), non-negative least square (NNLS), and expectation-maximization (EM) (Tsui et al., 1991; Lawson and Janson, 1995; Todd and Ramachandran, 1994). The NNLS algorithm has similar performance to the ART algorithm but shorter computation time (Hashmonay et al., 1999). It has been used in US EPA OTM-10 for horizontal radial plume mapping of air contaminants (EPA, 2005). The EM algorithm is mainly used for ML-based minimization. These

traditional pixel-based algorithms are suitable for rapid CT, but they produce maps with poor spatial resolution, owing to the requirement that the pixel number must not exceed the beam number, or they may have problem of indeterminacy associated with substantially underdetermined systems (Hashmonay, 2012).

To mitigate the problem of indeterminacy and improve the spatial resolution of reconstructions without substantially increasing the system cost, the smooth basis function minimization (SBFM) algorithm has been proposed. This algorithm represents the distribution map by a linear combination of several bivariate Gaussian functions (Drescher et al., 1996; Giuli et al., 1999). Each bivariate Gaussian has six unknown parameters (normalizing coefficient, correlation coefficient, peak locations and standard deviations) to be determined. The problem requires fitting these parameters to the observed PIC data. This method

performs better than the traditional pixel-based algorithms for ORS-CT applications because the patterns of air dispersion are physically smooth in shape (Wu and Chang, 2011). However, the resultant equations defined by the PICs are non-linear because of the unknown parameters. The search for the best-fit set of parameters minimizing the mean-squared difference between predicted and measured path integrals can be performed through an iterative minimization procedure, such as the simplex method or simulated annealing.

The reported methods using simulated annealing to find a global minimization are highly computationally intensive, thereby limiting the SBFM algorithm's practical applications, such as rapid reconstruction in industrial monitoring of chemical plants. However, an algorithm converging toward a smooth concentration distribution consistent with the path-integrated data has been demonstrated to be a rational choice. To improve the computational speed and append the smoothness *a priori* information to the inverse problem, the pixel-based low third derivative (LTD) algorithm has been proposed. This algorithm sets the third

derivative at each pixel to zero, thus resulting in a new system of linear equations that is overdetermined. The LTD algorithm has been reported to work as well as the SBFM algorithm, but is approximately 100 times faster (Price et al., 2001). Another method to produce the smoothness effect is the grid translation (GT) algorithm, which shifts the basis grid by different distances (e.g., 1/3 or 2/3 the width of the basis grid) horizontally and vertically while keeping the basis grid fixed (Verkruysse and Todd, 2004). Smoothness is achieved by averaging the reconstruction results after each shifting. An improved version called

grid-translation and multi-grid (GT-MG) applies the GT algorithm at different basis grid resolutions (Verkruysse and Todd, 2005). This method has been used with the ML-EM algorithm to improve the reconstruction accuracy, particularly in determining the peak location and value (Cehlin, 2019).

The success of these algorithms demonstrates the need to apply smoothness restriction to the ORS-CT gas mapping. With the LTD algorithm, a smooth reconstruction is achieved by simply adding the third-order derivative constraints. The generated

solutions are locally quadratic. To understand the characteristics of these constraints and apply the method to specific application, the theoretical basis of the algorithm must be understood. However, this basis is not clearly defined in the literature. With the purpose of introducing smoothness constraints, the LTD algorithm can be treated as a special case of the Tikhonov regularization, a well-known technique to solve the ill-posed inverse problem (Tikhonov and Arsenin, 1977; Rudin et al., 1992). The Tikhonov $L_2$ regularization uses a penalty term defined by the squared norm of the *ith*-order derivative of the

function and produces a smoothing effect on the resulting solution (Gholami and Hosseini, 2013). The third-order derivative is used in the LTD algorithm, although the first, second and higher order derivatives can also produce smooth results. A more flexible method of regularization uses the smoothness seminorm according to the variational interpolation theory, given its similar formula (Mitasova et al., 1995). The variational method is another way of achieving spline interpolation, given that the interpolation polynomial splines can be derived as the solution of certain variational problems of minimizing an integral whose

integrant consists of different order derivatives or their combinations.

The interpolation techniques are based on the given sample points, in contrast to tomographic reconstruction, in which only the line integrals are known. However, we have found that the interpolation can be adopted in the reconstruction process to produce a smooth solution by using the smoothness seminorm for interpolation as a smoothness regularization factor for the tomographic reconstruction problem. In view of variational spline interpolation, the characteristics of algorithms using

different seminorms have been well explored in the literature. The LTD algorithm can be considered as one case that minimizes the seminorm consisting of the third-order derivatives (Bini and Capovani, 1986). Other algorithms can also be formulated by using different seminorms. On the basis of this idea, we propose a new minimum curvature (MC) algorithm using a seminorm approximating the integral of the squares of the curvature. This algorithm generates a smooth reconstruction approximating the application of cubic spline interpolation. We compared the algorithm with the NNLS, LTD, and GT-MG algorithms by

using multiple test maps. We demonstrated its effectiveness and two main aspects of this method. First, smooth effect similar to spline interpolation is achieved during the reconstruction process by using high-resolution grid division, and second, the computational efficiency is markedly better than that of the LTD algorithm through halving the number of linear equations according to the new smoothness seminorm. This approach achieves the same performance but is easier to perform than the GT-MG algorithm which has complicated operations involving multiple grids and grid translation. More specific algorithms

applied for the ORS-CT method for mapping atmospheric chemicals could be further formulated and evaluated similarly.

## 2 Materials and methods

### 2.1 ORS-CT and beam geometry

The area of the test field was 40 m×40 m. Open-path TDL was used as the ORS analyzer, which was installed on a scanner and aimed at multiple retroreflectors by scanning periodically and continuously. To compare the results with those of the GT-

MG algorithm, we used an overlapping beam configuration similar to that used by Verkruysse and Todd (2005). As shown in Fig. 1, four TDL analyzers were located at the four corners of the test field. The retroreflectors were evenly distributed along the edges of the field. The total number of retroreflectors was 20. Each retroreflector reflected the laser beams coming from two different directions. Excluding the overlapped beams along the diagonals, the total beam number was 38. For the traditional pixel-based algorithm, the pixel number should not exceed the beam number. Therefore, we divided the test field into 6×6=36

pixels. The concentration within each pixel was assumed to be uniform.

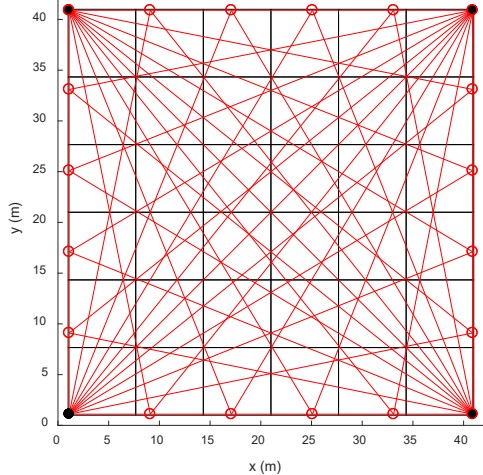

**Figure 1.** The beam configuration and grid division. The field was divided into 6×6 grid pixels. Four open-path TDL analyzers were located at the four corners. A total of 20 retroreflectors were distributed on the edges of the field.

The wavelength of the laser beam is tuned to the absorption line of the target gas and is transparent to other species. For general detection of atmospheric pollutants, the laser absorption is in the linear regime and the attenuation of the laser beam is governed by the Beer–Lambert law. The predicted PIC for one beam is equal to the sum of the multiplication of the pixel concentration and the length of the beam inside the pixel. In general, let us assume that the site is divided into $N_c = m \times n$ pixels, which are arranged as a vector according to the left-to-right and top-to-bottom sequence and indexed by $j$. The average concentration for the $j$-th pixel is $c_j$. The total number of laser beams is $N_b$, which are indexed by $i$. The length of the $i$-th laser beam passing the $j$-th pixel is $L_{ij}$. Then, for the $i$-th beam, the measured PIC $b_i$ is contributed by all pixels. We have the following linear equation

$$b_i = \sum_{j=1}^{N_c} L_{ij} c_j \tag{1}$$

A system of linear equations can be set up for all beams

$$\boldsymbol{b} = \boldsymbol{L}\boldsymbol{c} \tag{2}$$

where $\boldsymbol{L}$ is the kernel matrix that incorporates the specific beam geometry with the pixel dimensions, $\boldsymbol{c}$ is the unknown concentration vector of the pixels, and $\boldsymbol{b}$ is a vector of the measured PIC data. With the least squares approach, the reconstruction can be solved by minimizing the following problem

$$\min_{\boldsymbol{c}} \ \|\boldsymbol{L}\boldsymbol{c} - \boldsymbol{b}\|_2^2, \text{ subject to } \boldsymbol{c} \geq 0 \tag{3}$$

where $\|\cdot\|_2$ denotes the Euclidean norm. The non-negative constraints are applied to eliminate unrealistic negative resolutions resulting from observation error and ill-posed problem. This non-negative constrained linear least squares problem can be solved by the widely used NNLS optimization algorithm (Lawson and Janson, 1995), which is an active-set optimization method using an iterative procedure to converge on the best fit of positive values. The routine "lsqnonneg" in MATLAB software was used in this study. The optimal least squares solution is not smooth because the minimizing process does not

introduce smoothness *a priori* information. Herein, the NNLS algorithm in the tomographic reconstruction refers to solving the original problem by using the NNLS optimization algorithm without adding additional *a priori* information. When the system of linear equations is underdetermined, the solution is not unique. Additional information must be introduced to choose the appropriate solution.

## 2.2 LTD algorithm and Tikhonov regularization

The LTD algorithm introduces the smoothness information through setting the third-order derivative of the concentration to zero at each pixel in both $x$ and $y$ directions, thus generating solutions that are locally quadratic (Price et al., 2001). We define $c_j$ as an element of a one-dimensional (1-D) concentration vector of the pixels, but the pixels also have two-dimensional (2-D) structure according to the grid division of the site area and can be indexed by the row number $k$ and column number $l$, where $j=(k-1)n+l$. We use $C_{k,l}$ to denote the pixel concentration at the pixel located at the $k$-th row and $l$-th column of the grids. The third-derivative prior equations at the $(k, l)$ pixel are defined as

$$\frac{d^3C}{dx^3} = (C_{k+2,l} - 3C_{k+1,l} + 3C_{k,l} - C_{k-1,l})\frac{1}{(\Delta x)^3} = 0$$

$$\frac{d^3C}{dy^3} = (C_{k,l+2} - 3C_{k,l+1} + 3C_{k,l} - C_{k,l-1})\frac{1}{(\Delta y)^3} = 0 \tag{4}$$

where $\Delta x=\Delta y=\Delta d$ is the grid length in the $x$, $y$ direction. Therefore, two additional linear equations are introduced at each pixel defined by Eq. (4). There will be $2N_c$ linear equations appended to the original linear equations defined by Eq. (2), thus resulting in a new over-determined system of linear equations with $(2N_c +N_b)$ equations and $N_c$ unknowns.

A weight needs to be assigned to each equation depending on the uncertainty of the observation. Under the assumption that the analyzers have the same performance, the uncertainty is mainly associated with the path length. Therefore, equations are assigned weights inversely proportional to the path length to ensure that different paths have equal influences. Herein, the lengths of the laser paths are approximately equal to each other. Therefore, their weights are set to the same value and scaled to be 1. The weights for the third-derivative prior equations are assigned as the same value of $w$, because they are all based on the same grid length. The determination of $w$ follows the scheme for determining the regularization parameter described below. With the least squares approach, the reconstruction is intended to minimize the following problem

$$\min_{c}\left\|\begin{bmatrix} L \\ wT \end{bmatrix}c - \begin{bmatrix} b \\ 0 \end{bmatrix}\right\|_2^2, \text{ subject to } c \geq 0 \tag{5}$$

where $T$ is the kernel matrix for the third-derivative prior equations. Assuming that the new augmented kernel matrix is $A$ and the observation vector is $p$, the new system of linear equations will be $Ac=p$. The non-negative least squares solution was also found by the NNLS optimization algorithm. If the non-negative constraints are ignored, the least squares solution can be found analytically as $(A^TWA)^{-1}A^TWp$, where $W$ is a diagonal matrix whose diagonal elements are the weights (Price et al., 2001). However, this analytical solution may produce unrealistic large negative values and cannot be used in this study.

The LTD algorithm actually constructs a regularized inverse problem. It can be viewed as a special case of the well-known Tikhonov regularization technique. The Tikhonov $L_2$ regularization can be written as the following minimization problem (Gholami and Hosseini, 2013)

$$\min_{c} \quad \|\boldsymbol{Lc} - \boldsymbol{b}\|_2^2 + \mu\|\boldsymbol{D}_k\boldsymbol{c}\|_2^2 \tag{6}$$

where the first term represents the discrepancy between the measured and predicted values, the second term is the regularization term adding a smoothness penalty to the solution, $\mu$ is the regularization parameter controlling the conditioning of the problem, and matrix $\boldsymbol{D}_k$ is the regularization operator, which is typically a $k$th-order difference operator. The first- and second-order difference operators are commonly used. We can see that the LTD algorithm uses the third-order forward difference operator

$$\boldsymbol{D}_3 = \begin{bmatrix} -1 & 3 & -3 & 1 & & & \\ & -1 & 3 & -3 & 1 & & \\ & & & \ddots & & & \\ & & 1 & 3 & -3 & 1 & \\ & & & 1 & 3 & -3 & 1 \end{bmatrix} \frac{1}{\Delta d} \in \mathbb{R}^{(m-3)\times n} \tag{7}$$

For pixels on the edges, the second-order and first-order difference operators [1 -2 1] and [1 -1] can be used. The regularization parameter is analogous to the weight parameter for the prior equations in the LTD algorithm.

The regularization parameter determines the balance between data fidelity and regularization terms. Determination of the optimum regularization parameter is an important step in the regularization method. However, the regularization parameter is problem and data dependent. There is no general-purpose parameter-choice algorithm that will always produce a good parameter. For simplicity, we use the method based on the discrepancy principle (Hamarik et al., 2012). The regularization parameter $\mu$ is chosen from a finite section of a monotonic sequence. For each value of $\mu$, an optimal solution is derived by solving the inverse problem. The discrepancy can then be calculated. The regularization parameter is determined to be the highest value that makes the discrepancy $\|\boldsymbol{Lc} - \boldsymbol{b}\|_2^2$ equal to $N_b\sigma^2$, where $\sigma$ is the standard deviation of the noise. In this study, the reconstructions varied only slowly with the regularization parameters. Therefore, precise selection of the parameter was not necessary. For computational efficiency, the regularization parameter was selected from four widely varying values. The one producing the smallest discrepancy was used.

**2.3 Variational interpolation and minimum curvature algorithm**

Splines are special types of piecewise polynomials, which have been demonstrated to be very useful in numerical analysis and in many applications in science and engineering problems. They match given values at some points (called knots) and have continuous derivatives up to some order at these points (Champion et al., 2000). Spline interpolation is preferred over polynomial interpolation by fitting low-degree polynomials between each of the pairs of the data points instead of fitting a single high-degree polynomial. Normally, the spline functions can be found by solving a system of linear equations with unknown coefficients of the low-degree polynomials defined by the given boundary conditions.

The variational approach provides a new way to find the interpolating splines and opens up directions in theoretical developments and new applications (Champion et al., 2000). Variational interpolation was motivated by the minimum curvature property of natural cubic splines, i.e., the interpolated surface minimizes an energy functional that corresponds to the potential energy stored in a bended elastic object. This principle provides flexibility in controlling the behavior of the generated spline. Given an observation $z_k$ ($k=1, \ldots, N$) measured at the $k$-th point whose position vector is $\mathbf{r}_k$, a spline function

$F(\mathbf{r})$ for interpolating the data points can be found through the variational approach by minimizing the sum of the deviation from the measured points and the smoothness seminorm of the spline function

$$\min_{F} \ \sum_{k=1}^{N} |F(\mathbf{r}_k) - z_k|^2 + \mu I(F) \tag{8}$$

where $\mu$ is a positive weight, and $I(F)$ denotes the smoothness seminorm. The seminorm can be defined in various forms, commonly the first, second, third derivatives, or their combinations. The solution to the minimizing problem is spline functions,

which can also be found by solving a Euler-Lagrange differential equation corresponding to the given seminorm (Briggs, 1974).

We can see that the minimizing problem in Eq. (8) has a similar form to the Tikhonov regularization but with a more flexible regularization term. The problem is that the variational interpolation is based on given data points, whereas the tomographic reconstruction is based on measured line integrals. However, we show herein that the variational approach for interpolation

can also be applied to the latter problem to produce a smoothness solution with an effect similar to spline interpolation. In addition, on the basis of different seminorms, we can formulate many different reconstruction algorithms. In this way, we propose a new minimum curvature (MC) algorithm.

Under the assumption that the unknown concentration distribution is described by a function $f(x, y)$, $(x_k, y_l)$ are the smallest coordinates of the $j$-th pixel at row $k$ and column $l$ of the 2-D grids, then the concentration $c_j$ equals the average concentration

of the pixel

$$c_j = \frac{1}{(\Delta d)^2} \int_{x_k}^{x_{k+1}} \int_{y_l}^{y_{l+1}} f(x,y) dx dy \tag{9}$$

The minimization problem according to the variational approach is formulated as

$$\min_{f} \ \sum_{i=1}^{N_b} \sum_{j=1}^{N_c} \| L_{ij} c_j - b_i \|_2^2 + \mu I(f) \tag{10}$$

For the MC algorithm, we define the seminorm according to the minimum curvature principle, which is used in the geographic

data interpolation to seek a 2-D surface with continuous second derivatives and minimal total squared curvature (Briggs, 1974). The minimum-curvature surface is analogous to elastic plate flexure, and it approximates the shape adopted by a thin plate flexed to pass through the observation data points with a minimum amount of bending. This method generates the smoothest possible surface while attempting to follow the observation data as closely as possible. The seminorm in the MC algorithm is defined to be equal to the total squares curvature

$$I(f) \ \ = \int \int \left( \frac{\partial^2 f}{\partial x^2} + \frac{\partial^2 f}{\partial y^2} \right)^2 dx dy \tag{11}$$

This integral must be discretized according to the grid division. The discrete total squares curvature is

$$S(C) = \sum_{k=1}^{n} \sum_{l=1}^{m} (I_{k,l})^2 (\Delta d)^2$$

(12)

where $I_{k,l}$ is the curvature at the $(k,l)$ pixel, which is a function of $C_{k,l}$ and its neighboring pixel values. In two dimensions the approximation to the curvature is

$$I_{k,l} = (C_{k+1,l} + C_{k-1,l} + C_{k,l+1} + C_{k,l-1} - 4C_{k,l})/(\Delta d)^2$$

(13)

To minimize the total squared curvature, we need

$$\frac{\partial S}{\partial c_{k,l}} = 0$$

(14)

Combining Eq. (11), (12), and (13), we obtain the following difference equation

$$\begin{aligned}[C_{k+2,l} + C_{k,l+2} + C_{k-2,l} + C_{k,l-2} \\ + 2(C_{k+1,l+1} + C_{k-1,l+1} + C_{k+1,l-1} + C_{k-1,l-1}) \\ - 8(C_{k+1,l} + C_{k-1,l} + C_{k,l-1} + C_{k,l+1}) + 20C_{k,l}]/(\Delta d)^2 = 0\end{aligned}$$

(15)

This equation is appended at each pixel as a smoothness regularization. Therefore, there is only one prior equation at each grid instead of two equations in the LTD algorithm. For pixels on the edges, we set the approximation of the first and second derivatives to zeros by using the difference operators [1 -1] and [1 -2 1]. Under the assumption that $M$ is the kernel matrix of the prior equations, the reconstruction aims to minimize the following problem

$$\min_{c} \quad \|Lc - b\|_2^2 + \mu \|Mc\|_2^2, \text{ subject to } c \geq 0$$

(16)

where the parameter $\mu$ is determined in the same manner as the regularization parameter in Tikhonov regularization method. Similar to the LTD approach, the resulting constrained system of linear equations is over-determined and is solved by the NNLS optimization algorithm.

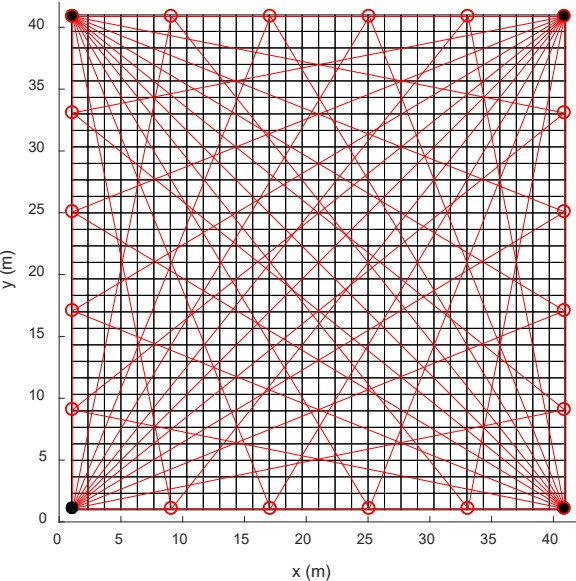

**Figure 2.** Beam geometry and a 30×30 grid division of the site.

For conventional pixel-based reconstruction algorithms, the number of pixels (unknowns) should not exceed the number of beams (equations) to obtain a well-posed problem. Because only tens of beams are usually used in ORS-CT applications, the resultant spatial resolution is very coarse. The GT algorithm is one way to increase the resolution, but it requires several steps to complete the entire translation because each translation uses a different grid division, and the reconstruction process must be conducted for each grid division. In the MC algorithm, we use only one division of high-resolution grids directly during the reconstruction. The resultant system of linear equations remains determined because of the smoothness restriction at each pixel. As shown in Fig. 2, 30×30 pixels are used in the MC algorithm instead of the 6×6 pixels in the NNLS approach. Under this configuration, the number of linear equations for the LTD algorithm is approximately 38+30×30×2=1838, whereas the number for the MC algorithm is approximately 38+30×30=938. Thus, the MC approach decreases the number of linear equations to approximately half that of the LTD algorithm. The smoothness seminorm of the MC algorithm ensures a smooth solution. This smoothing effect is similar to the spline interpolation applied after the reconstruction process, except that it is achieved during the inverse process. This aspect is important because an interpolation after the reconstruction cannot correct the error resulting from the reconstruction in terms of coarse spatial resolution. The MC approach evaluates the discrepancy based on the high-resolution values that are the same as the reconstruction outcomes. Errors due to coarse spatial resolution are corrected during the process.

**2.4 Test concentration data**

The NNLS, LTD, and MC algorithms were compared by using multiple test maps. The results were also compared with those of the GT-MG algorithm. We set up test conditions similar to those used in Verkruysse and Todd (2005). The concentration distribution from one source is defined by a bivariate Gaussian distribution

$$g(x,y) = Q\exp[-\left(\frac{(x-x_0)^2}{\sigma_x^2} + \frac{(y-y_0)^2}{\sigma_y^2}\right)] \tag{17}$$

where $Q$ (0 to 40 mg/m$^3$) is the source strength, $x_0$, $y_0$ (0 to 40 m) is the peak location, and $\sigma_e$, $\sigma_y$ are the width of the peaks with possible values of 2.8, 4.2, 5.7, and 7.1 m.

The source number varies from 1 to 5. For multiple sources, the resultant concentration distribution is the superposition value due to each source. For each source number, 100 maps were generated by randomly setting the source strength, location, and
280 peak width from the defined ranges or set above.

The concentration filed is discretized with a resolution of 0.2 m×0.2 m. The concentration of each pixel is the average value of the concentrations in that pixel. The discretized concentration map is used as the true concentration distribution. PICs are calculated based on the discretized map by using Eq. (1).

**2.5 Evaluation of reconstruction quality**

A conventional image quality measure called nearness is used to describe the discrepancy between the original maps and the reconstructed maps. Nearness evaluates errors over all grid cells on the map (Verkruysse and Todd, 2005)

$$\text{Nearness} = \sqrt{\frac{\sum_i^{m\times n}(c_i^* - c_i)^2}{\sum_i^{m\times n}(c_i^* - c_{avg}^*)^2}} \tag{18}$$

where $m$, $n$ are the grid divisions on the $x$, $y$ direction of the map, $c_i^*$ is the synthetic value of concentration in the $i$th grid generated by the Gaussian distribution model, $c_i$ is the estimated value for the $i$th grid, and $c_{avg}^*$ is the mean concentration of
290 all grids. A nearness value of zero implies a perfect match.

The effectiveness of locating the emission source is evaluated by the peak location error, which calculates the distance between the true and reconstructed peak locations.

$$\text{Peak location error} = \sqrt{(x_r - x_0)^2 + (y_r - y_0)^2} \tag{19}$$

where $x_r$, $y_r$ are the peak locations on the reconstruction map. For multiple peaks, only the location of the highest peak is
295 calculated. The peak is located by searching for the largest concentration on the map. When multiple locations have the same values, the centroid of these locations is used.

Exposure error percentage is used to evaluate how well average concentrations in the whole field are reconstructed. It can reflect the accuracy of measuring chemical air emissions and emission rates from fugitive sources, such as agricultural sources and landfills (Verkruysse and Todd, 2004)

$$\text{Exposure error \%} = \left| \frac{\Sigma_i^{m \times n} c_i^* - \Sigma_i^{m \times n} c_i}{\Sigma_i^{m \times n} c_i^*} \right| \times 100\% \tag{20}$$

Herein, a measure using the averaging kernel matrix is also applied to predict the reconstruction error due to different regularization approaches. Resolution matrices are commonly used to determine whether model parameters can be independently predicted or resolved, and how regularization limits reconstruction accuracy (Twynstra and Daun, 2012; von Clarmann et al., 2009). Ignoring the non-negative constraints, the generalized inverse matrices for the NNLS, LTD, and MC algorithms can be found by

$$\boldsymbol{G}_{NNLS} = (\boldsymbol{L}^T \boldsymbol{L})^{-1} \boldsymbol{L}^T$$

$$\boldsymbol{G}_{LTD} = (\boldsymbol{L}^T \boldsymbol{L} + \mu^2 \boldsymbol{D}_3^T \boldsymbol{D}_3)^{-1} \boldsymbol{L}^T$$

$$\boldsymbol{G}_{MC} = (\boldsymbol{L}^T \boldsymbol{L} + \lambda^2 \boldsymbol{M}^T \boldsymbol{M})^{-1} \boldsymbol{L}^T \tag{21}$$

The averaging kernel matrix is defined as $\boldsymbol{R} = \boldsymbol{GL}$. The reconstruction error is given by

$$\delta \boldsymbol{c} = \boldsymbol{c}_{model} - \boldsymbol{c}_{exact} = (\boldsymbol{R} - \boldsymbol{I}) \boldsymbol{c}_{exact} - \boldsymbol{G} \delta \boldsymbol{b} \tag{22}$$

where $\boldsymbol{c}_{model}$ and $\boldsymbol{c}_{exact}$ are the model-predicted and the exact concentrations, respectively, $\delta \boldsymbol{b}$ is the perturbation of the observation resulting from various noise sources, $\boldsymbol{I}$ is the identity matrix, $(\boldsymbol{R} - \boldsymbol{I}) \boldsymbol{c}_{exact}$ is the regularization error caused by the inconsistency between the measurement data equations and the prior information equations, and $\boldsymbol{G} \delta \boldsymbol{b}$ is the perturbation error.

For the LTD and MC approaches using high-resolution grids, the kernel matrix $\boldsymbol{L}$ is rank-deficient, and the regularized solution is robust to perturbation error over a wide range of regularization parameters. Thus, the perturbation error is negligible, and the reconstruction error is dominated by regularization error (Twynstra and Daun, 2012). Because the averaging kernel matrix is determined only by the beam configuration and the regularization approach, it is independent of the actual concentration distribution. Therefore, it is best used to evaluate different beam configurations that considerably influence the reconstruction accuracy. However, in this study the beam configurations are fixed. We can therefore use the averaging kernel matrix to measure different regularization approaches. In an ideal experiment, $\boldsymbol{R} = \boldsymbol{I}$, thus implying that each unknown pixel value can be independently resolved from the measurement data. The regularization term forces the off-diagonal terms in $\boldsymbol{R}$ to be nonzero, thereby making the estimated concentration of each pixel a weighted average of the concentration of the surrounding pixels. We can use the Frobenius norm (the square root of the sum of the absolute squares of the elements of a matrix) between $\boldsymbol{R}$ and $\boldsymbol{I}$ defining a measure of fitness to predict the reconstruction error (Twynstra and Daun, 2012).

$$\varepsilon = \frac{1}{N_c} \|\boldsymbol{R} - \boldsymbol{I}\|_F^2 \tag{23}$$

## 3 Results and discussions

In these tests, the traditional NNLS algorithm uses 6×6 grids, whereas the LTD and MC algorithms both use 30×30 grids. The results of the GT-MG algorithm are from Verkruysse and Todd (2005), in which a maximum basis grid resolution of 10×10

with 1/4 grid size as translation distance was used. Of note, the test conditions were not exactly the same as those used by the GT-MG algorithm, which did not measure the peak location error and used a different method to calculate the exposure error by limiting the calculation domain to a small area near the peak instead of the entire map. Therefore, the results of the GT-MG algorithm are provided as a reference and only the measure of nearness was compared. The original resolution of the reconstruction map by the NNLS algorithm is too coarse (6.7 m). To determine the peak locations more accurately, all

concentration maps reconstructed by the NNLS algorithm were spline interpolated with a resolution of 0.5 m. Fig. 3 depicts some examples of the test maps and reconstructed maps generated by different algorithms with different source numbers.

(a)

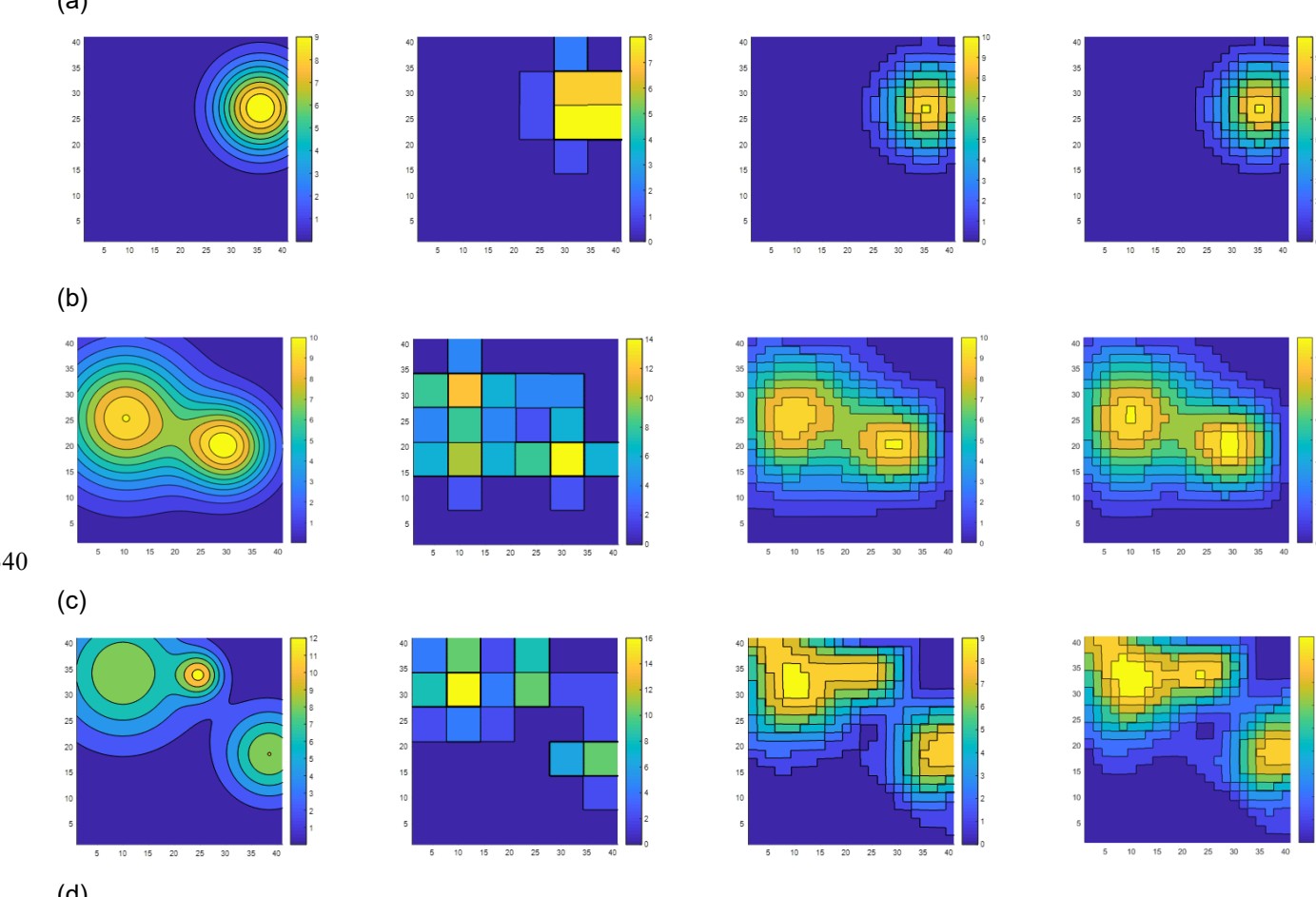


(b)

(c)

(d)

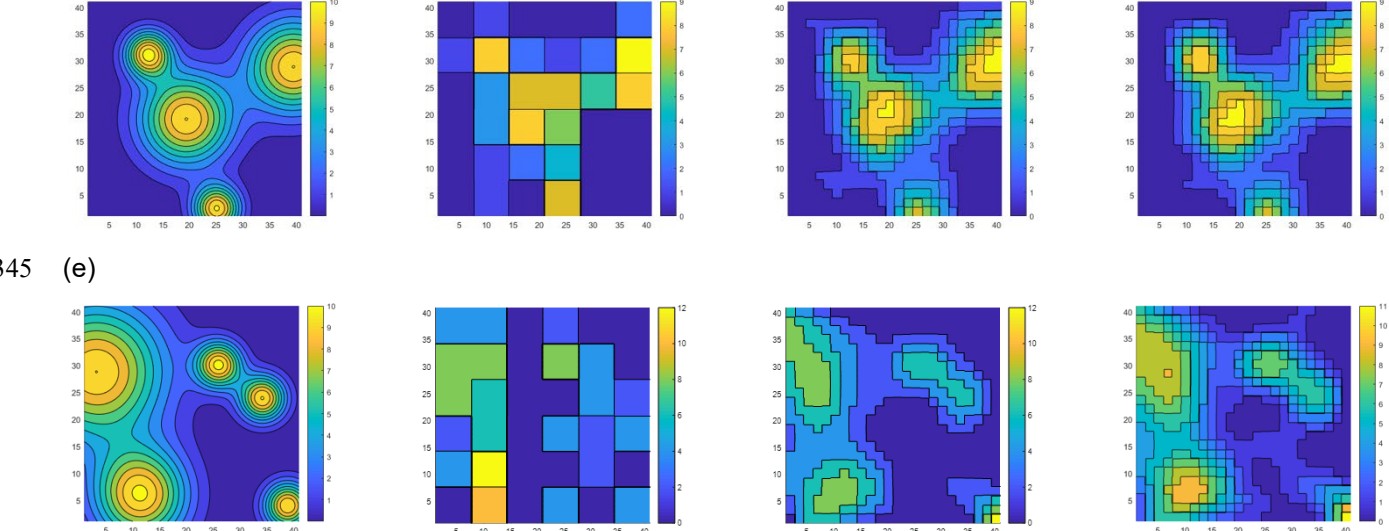

(e)

**Figure 3.** Original test maps (first column) and corresponding maps reconstructed with the NNLS (second column), LTD (third column), and MC (forth column) algorithms. (a) One source; (b) two sources; (c) three sources; (d) four sources; (e) five sources.

## 3.1 Nearness

**Table 1.** Mean and standard deviation of nearness.

| Source number | NNLS | LTD | MC | GT-MG[*] |
|---|---|---|---|---|
| 1 | 0.40 (0.21) | 0.13 (0.08) | 0.11 (0.07) | 0.09 (0.05) |
| 2 | 0.38 (0.16) | 0.15 (0.07) | 0.13 (0.06) | 0.16 (0.07) |
| 3 | 0.40 (0.14) | 0.18 (0.08) | 0.17 (0.08) | 0.19 (0.06) |
| 4 | 0.40 (0.12) | 0.20 (0.08) | 0.19 (0.08) | 0.25 (0.08) |
| 5 | 0.43 (0.13) | 0.22 (0.09) | 0.21 (0.08) | 0.27 (0.09) |

[*]: The results of the GT-MG algorithm are from Verkruysse and Todd (2005), whose test conditions are not exactly the same as the conditions used herein.

Nearness is the most important measure of accuracy of the reconstructed map. It represents the reconstruction of peak heights, shapes, and the production of artifacts. The smaller the nearness value, the better the reconstruction quality. In Table 1, the

LTD, MC and GT-MG algorithms generally reduce the nearness values by more than 50% with respect to the values obtained by the NNLS algorithm. Under the condition of one source, they reduce the nearness by approximately 70% with respect to the NNLS. The LTD, MC and GT-MG algorithms show increasing trends as the source number increases, thus implying that the performance of the algorithm is affected by the complexity of the underlying distribution. The nearness results of NNLS for different numbers of sources are almost the same because they are the results after spline interpolation. In fact, the original

un-interpolated results also show increasing trends. The interpolation improves the results of the NNLS algorithm more than

those of the LTD and MC algorithms, which already use high-resolution grids. The overall performance of the LTD, MC, and GT-MG algorithms is very similar, whereas the new MC algorithm's performance is slightly better.

## 3.2 Peak location error

**Table 2.** Mean and standard deviation of peak location error.

| Source number | NNLS (m) | LTD (m) | MC (m) |
| --- | --- | --- | --- |
| 1 | 1.78 (0.93) | 0.41 (0.45) | 0.40 (0.56) |
| 2 | 4.88 (8.21) | 1.97 (5.98) | 1.62 (4.81) |
| 3 | 5.17 (8.35) | 2.58 (6.77) | 2.34 (6.17) |
| 4 | 8.40 (11.53) | 5.22 (10.28) | 5.58 (10.76) |
| 5 | 8.95 (11.32) | 5.51 (10.15) | 5.77 (10.41) |

As shown in Table 2, the LTD and MC algorithms show better performance in peak location error than the NNLS algorithm. They generally improve the accuracy of peak location by 1 to 2 m. The errors of all algorithms increase with the source number. One reason for this finding is that when two or more peaks with comparable peak magnitudes on the map (Fig. 3), the algorithm may not identify the correct location of the highest peak. Therefore, a large error may occur when the highest value on the reconstructed map is located on the wrong peak.

## 3.3 Exposure error

**Table 3.** Mean and standard deviation of exposure error.

| Source number | NNLS (%) | LTD (%) | MC (%) |
| --- | --- | --- | --- |
| 1 | 5.18 (7.69) | 1.30 (1.51) | 1.04 (0.85) |
| 2 | 3.03 (3.68) | 1.11 (0.97) | 1.07 (0.92) |
| 3 | 2.74 (2.90) | 1.16 (0.84) | 1.11 (0.76) |
| 4 | 2.29 (2.09) | 1.21 (1.04) | 1.12 (0.98) |
| 5 | 2.26 (1.72) | 1.18 (0.87) | 1.16 (0.84) |

The exposure error of NNLS can be severely affected by the spline interpolation of the reconstruction results. Therefore, a nearest interpolation was used. As shown in Table 3, MC and LTD algorithms show approximately 2 to 5 times better performance than the NNLS algorithm. The exposure error reflects the accuracy of the overall emissions measurement other 375 than the concentration distribution. The performance of the LTD and MC algorithms is very similar, whereas the MC algorithm illustrates slightly better performance than the LTD algorithm. Unlike the trends shown by the NNLS in the nearness and peak location error, its performance in exposure error improves with increasing source number. A plausible cause of this phenomenon may be that the distribution becomes more uniform with larger numbers of sources. Because the NNLS algorithm

uses coarse grid division, it produces concentrations with very low spatial resolution and fits the true distribution better when
the distribution becomes more uniform.

## 3.4 Computation time

**Table 4.** Mean and standard derivation of computation time.

| Source number | LTD (s) | MC (s) | Ratio (MC/LTD) |
|:---:|:---:|:---:|:---:|
| 1 | 11.08 (14.27) | 8.06 (10.03) | 0.73 |
| 2 | 21.02 (17.89) | 14.17 (11.46) | 0.67 |
| 3 | 34.44 (19.93) | 22.76 (13.61) | 0.66 |
| 4 | 43.58 (20.45) | 29.13 (12.92) | 0.67 |
| 5 | 59.48 (21.98) | 38.74 (15.05) | 0.65 |

The computations were run on a computer with a processor of Intel Core i7-6600U 2.6 GHz and RAM of 8 GB. In Table 4,
the computation times for the LTD and MC algorithms are compared. The computation time generally increases with
increasing source number. The MC algorithm is faster than the LTD algorithm because it has approximately half the number
of the linear equations as the LTD algorithm. The ratio results show that the MC algorithm's computation time is approximately
65% that of the LTD algorithm when the source number is five. The trend of the ratio implies that the advantage of the MC
algorithm becomes more clearer with increasing complexity of the underlying distribution.

## 3.5 Fitness

Contour plots of the resolution matrices for the LTD and MC algorithms are shown in Fig. 4. (a) and (b). Each row represents
the weight strength of all the pixels for the current pixel. The fitness values for the LTD and MC algorithm are 1.4411 and
1.3878, respectively. The MC algorithm shows slightly better performance. The off-diagonal elements are not zeros. The
reconstructed concentration at each pixel is a weighted average of the concentrations of the surrounding pixels according to
the smoothness regularization. Each row of the averaging kernel matrix can be regarded as smoothing weights. Because the
pixels have a 2-D arrangement, we show the 2-D display of the row of the 106th pixel (row and column indices are 4 and 16)
in the averaging kernel matrix for the LTD and MC algorithms in Fig. 4. (c) and (d) as an example. The dependence on the
beam geometry can be seen on both pictures. Because the beam configuration is fixed, the difference between the fitness values
is mainly caused by the use of different regularization approaches. The fitness difference between the LTD and MC algorithms
is very small, which may indicate that both algorithms have similar smoothness effects. This result coincides with the results
from other measures discussed above. The 2-D display of the diagonal elements of the averaging kernel matrix are shown in
Fig. 4. (e) and (f), which are not much useful in this case.

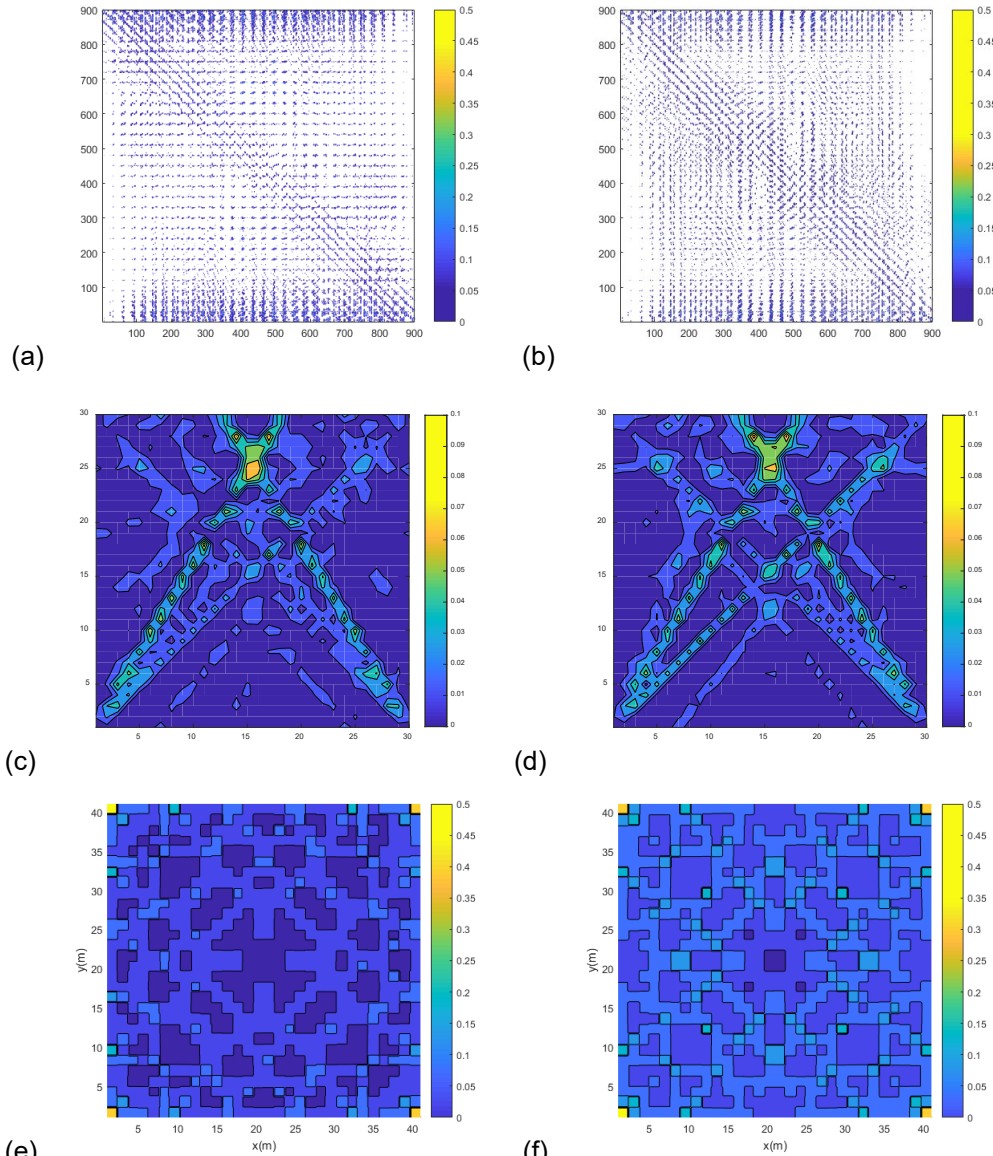

 **Figure 4.** Contour plot of the averaging kernel matrix for (a) the LTD algorithm (b) the MC algorithm. 2-D display of the row vector of the 106th pixel in the averaging kernel matrix for (c) the LTD algorithm (d) the MC algorithm. 2-D display of the diagonal elements of the averaging kernel matrix for (e) the LTD algorithm (f) the MC algorithm.

### 3.6 Influence of the grid size

The derivatives are approximated by the finite differences during the discretization process. The finite grid length causes

discretization error and affects the reconstruction results. We studied the influences of different grid divisions by investigating

the changes of the nearness, peak location error, exposure error, and computation time with respect to the pixel number. Five

different grid divisions were used: 6×6, 12×12, 18×18, 24×24, and 30×30. The peak number was five. A total of 100 maps were tested for each grid division. The results of the averaged values are shown in Fig. 5.

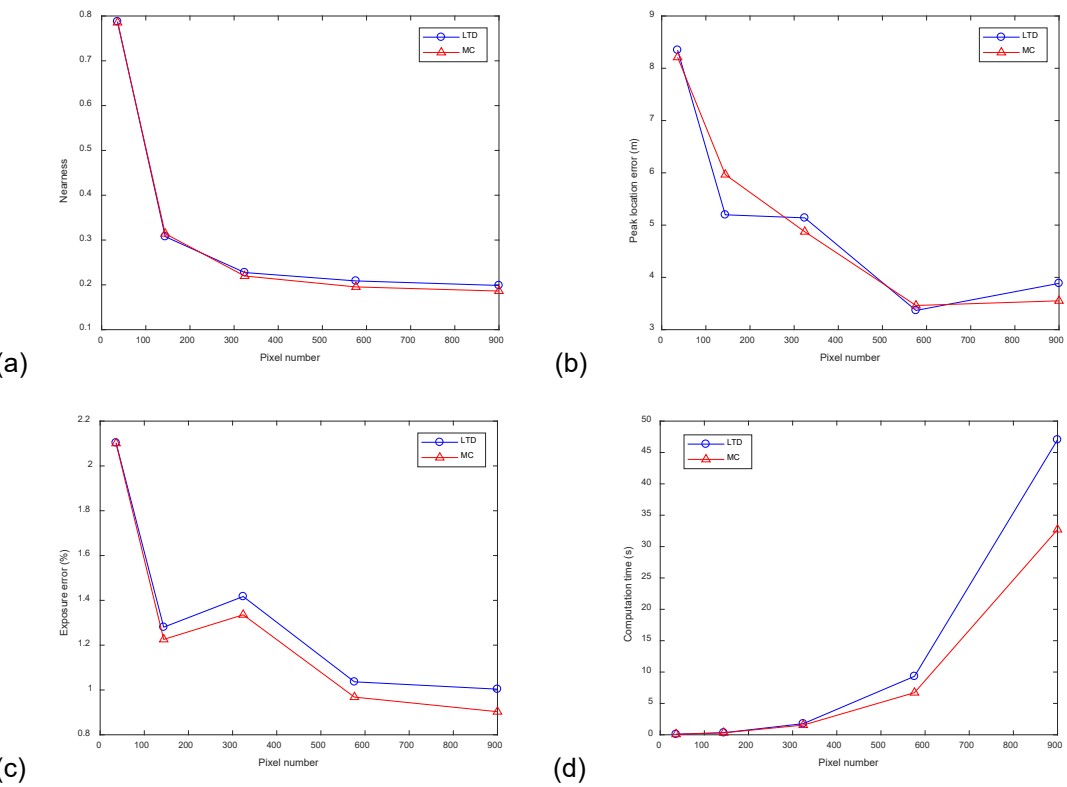

**Figure 5**. The change of (a) nearness (b) peak location error (c) exposure error percentage (d) computation time with respect to the pixel number.

The nearness, peak location error, and exposure error generally illustrate decreasing trends with increasing pixel number. The MC algorithm shows slightly better performance than the LTD algorithm with increasing pixel number. The performance improvement becomes slow for both algorithms when the division is finer than 24×24. The computation time shows approximately exponential growth trend with increasing pixel number. The LTD algorithm has a faster increasing rate than the MC algorithm. To conclude, the reconstruction performance is improved for both LTD and MC algorithms with increasing pixel numbers, but at the cost of fast growth of the computation time. And the improvement becomes small when the resolution is higher than certain threshold value (24×24 herein). Therefore, there should be a balance between the performance and the computation time.

## 4 Conclusion

To understand the characteristics of the smoothness constraints and to seek more flexible smooth reconstruction, we first identified the LTD algorithm as a special case of Tikhonov regularization. Then, more flexible smoothness constraints were found through the smoothness seminorms according to variational interpolation theory. The smoothness seminorms were successfully adopted in ORS-CT inverse problems. On the basis of variational approach, we proposed a new MC algorithm by using a seminorm approximating the sum of the squares of the curvature. The new algorithm improves computational efficiency through reducing the number of linear equations to half that of the LTD algorithm. It is simpler to perform than the GT-MG algorithm by directly using high-resolution grids during the reconstruction.

The MC, LTD, and NNLS algorithms were compared by using multiple test maps. The new MC algorithm shows similar performance as the LTD algorithm, but only requires approximately 65% the computation time. The smoothness-related algorithms of LTD, MC, and GT-MG all show better performance than the traditional NNLS algorithm: the nearness of reconstructed maps is improved by more than 50%, the peak distance accuracy is improved by 1- 2 m, and the exposure error is improved by 2 to 5 times. Because differences in accuracy between the LTD and MC algorithms are very small, more specific evaluations may be needed by using more complicated and realistic conditions.

These comparisons demonstrate the feasibility of introducing the theory of variational interpolation. On the basis of the seminorms, it is easier to understand the advantages and the drawbacks of different algorithms. Common problems such as the over-smooth issue may be improved by formulating more algorithms suitable for ORS-CT applications. Note that although the smoothness is very good *a priori* information for the reconstruction problem, beam configuration and underlying concentration distribution are also important factors affecting the reconstruction quality. To further improve the reconstruction quality, extra *a priori* information according to specific application may be added to the inverse problem. For example, statistic information of the underlying distribution or information resulting from the fluid mechanics.

*Code and data availability*. Data and code are available on request by contacting the authors.

*Author contributions*. KD was responsible for acquiring funding for this research. SL designed the algorithm and conducted the tests. SL and KD were both involved in data analysis. Both authors contributed to writing and editing the manuscript.

*Competing interests*. The authors declare that they have no conflict of interest.

*Acknowledgements*. The authors are grateful to the supports by the following grants: Discovery Grant from Natural Sciences and Engineering Research Council (NSERC) of Canada (RGPIN-2020-05223), John R. Evans Leaders Fund (JELF) and Infrastructure Operating Fund (IOF) from Canada Foundation for Innovation (CFI) (35468), University Research Grant Committee (URGC) seed grant from University of Calgary (1050666).

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
