# Peer review of "A minimum curvature algorithm for tomographic reconstruction of atmospheric chemicals based on optical remote sensing"

_Atmospheric Measurement Techniques, 2021_

## Author Comment (AC3)

**Supplement**

…

**2 Materials and methodologies**

**2.1 ORS-CT and beam geometry**

5   The area of the test field is 40 m × 40 m. open-path TDL is used as the ORS analyzer, which is installed on a scanner and aims at multiple retroreflectors by scanning periodically and continuously. To compare with the results of GT-MG algorithm, we used an overlapping beam configuration similar to the one used in Verkruysse and Todd (2005). As shown in Fig. 1, four TDL analyzers are located at the four corners of the test field. The retroreflectors are evenly distributed along the edges of the field. The total number of retroreflectors is 20. Each retroreflector reflects the laser beams coming from two different directions.

10  Neglecting the overlapped beams along the diagonals, total beam number is 38. For traditional pixel-based algorithm, the pixel number should be no more than the beam number. Therefore, we divide the test field into 6 × 6 = 36 pixels. The concentration within each pixel is assumed to be uniform.

[Figure]

**Figure 1.** The beam configuration and grid division. The field is divided into 6 × 6 grid pixels. Four open-path TDL analyzers locate at four
15  corners. 20 retroreflectors are distributed on the edges of the field.

For each laser beam, the path-integrated concentration (PIC) is measured by the analyzer. The predicated PIC for one beam equals to the sum of the multiplication of the pixel concentration and the length of the beam inside the pixel. In generality, let us assume that the site is divided into $N_c=m×n$ pixels, which are arranged as a vector according to the left-to-right and top-to-bottom sequence and indexed by $j$. The average concentration for $j$-th pixel is $c_j$. The total number of laser beams is $N_b$ which

20   are indexed by $i$. The length of the $i$-th laser beam inside the $j$-th pixel is $L_{ij}$. Then for the $i$-th beam, the measured PIC $b_i$ is contributed by all the pixels. We have the following linear equation

$$b_i = \sum_{j=1}^{N_c} L_{ij} c_j \tag{1}$$

A system of linear equations can be set up for all the beams

$$\boldsymbol{b} = \boldsymbol{Lc} \tag{2}$$

25   where $\boldsymbol{L}$ is the kernel matrix that incorporates the specific beam geometry with the pixel dimensions, $\boldsymbol{c}$ is the unknown concentration vector of the pixels, $\boldsymbol{b}$ is a vector of the measured PIC data. Using least square approach, the reconstruction is to minimize the following problem

$$\min_{\boldsymbol{c}} \ ||\boldsymbol{Lc} - \boldsymbol{b}||_2^2, \text{ subject to } \boldsymbol{c} \geq 0 \tag{3}$$

where $||\cdot||_2$ denotes the Euclidean norm. This non-negative constrained linear least square problem can be solved by the widely

30   used NNLS optimization algorithm (Lawson and Janson, 1995), which is an active-set optimization method using an iterative procedure to converge to the best fit of positive values. The realization in MATLAB software as the routine "lsqnonneg" was used in this study. The optimal least square solution is not smooth because the minimizing process does not introduce smooth *a priori* information. In this paper, the "NNLS algorithm" to the tomographic reconstruction refers to solve the original problem using the NNLS optimization algorithm without adding additional *a priori* information. When the system of linear equations

35   is underdetermined, the solution is not unique. Additional information needs to be introduced to choose the appropriate solution.

**2.2 LTD algorithm and Tikhonov regularization**

The LTD algorithm introduces the smoothness *a priori* information through setting the third-order derivative of the concentration to be zero at each pixel in both $x$ and $y$ directions, which will generate solutions that are locally quadratic (Price

40   et al., 2001). We have defined $c_j$ as an element of one-dimensional (1-D) concentration vector of the pixels, but the pixels also have two-dimensional (2-D) structure according to the grid division of the site area and can be indexed by the row number $k$ and column number $l$, where $j=(k-1)n+l$. We use $C_{k,l}$ denotes the pixel concentration at pixel located at $k$-th row and $l$-th column of the grids. The third-derivative prior equations at $(k, l)$ pixel is define as

$$\frac{d^3 C}{dx^3} = (C_{k+2,l} - 3C_{k+1,l} + 3C_{k,l} - C_{k-1,l})\frac{1}{\Delta d} = 0$$

45
$$\frac{d^3 C}{dy^3} = (C_{k,l+2} - 3C_{k,l+1} + 3C_{k,l} - C_{k,l-1})\frac{1}{\Delta d} = 0 \tag{4}$$

where $\Delta d=\Delta x=\Delta y$ is the grid length in $x$, $y$ direction. Therefore, two additional linear equations are introduced at each pixel defined by Eq. (4). There will be $2N_c$ linear equations appended to the original linear equations defined by Eq. (2), resulting in a new over-determined system of linear equations which has $(2N_c +N_b)$ equations and $N_c$ unknowns.

A weight needs to be assigned to each equation depending on the uncertainty of the observation. Assuming the analyzers have

50   the same performance, the uncertainty is mainly related to the path length. Therefore, equations are assigned weights inversely

proportional to path length to make sure different paths have equal influences. In this study, the lengths of the laser paths are approximately equal to each other. So their weights were set to be the same value and scaled to be 1. The weights for the third-derivative prior equations were assigned to be the same value of $w$ because they were all based on the same grid length. The determination of $w$ follows the scheme of determining the regularization parameter described in the following text. Using least square approach, the reconstruction is to minimize the following problem

$$\min_{c}\left\|\begin{bmatrix} L \\ wT \end{bmatrix} c - \begin{bmatrix} b \\ 0 \end{bmatrix}\right\|_2^2, \text{ subject to } c \geq 0 \tag{5}$$

where $T$ is the kernel matrix for the third-derivative prior equations. Assuming the new augmented kernel matrix is $A$, observation vector is $p$, then the new system of linear equations is $Ac=p$. The non-negative least square solution was also found by the NNLS optimization algorithm. If the non-negative constrains are ignored, the least square solution can be found analytically as $\hat{c} = (A^T W A)^{-1} A^T W p$, where $W$ is a diagonal matrix whose diagonal elements are the weights (Price et al., 2001).

The process of the LTD algorithm actually constructs a regularized inverse problem. It can be viewed as one special case of the well-known Tikhonov regularization technique. The Tikhonov $L_2$ regularization can be written as the following minimization problem (Gholami and Hosseini, 2013)

$$\min_{c} \quad \|Lc - b\|_2^2 + \mu\|D_k c\|_2^2 \tag{6}$$

where the first term represents the discrepancy between the measured and predicated values, the second term is the regularization term adding a smoothness penalty to the solution, $\mu$ is the regularization parameter controlling the conditioning of the problem, matrix $D_k$ is the regularization operator, which is typically a $k$th-order difference operator. The first- and the second-order difference operators are commonly used. We can see that the LTD algorithm uses the third-order forward difference operator

$$D_3 = \begin{bmatrix} -1 & 3 & -3 & 1 & & & \\ & -1 & 3 & -3 & 1 & & \\ & & & \ddots & & & \\ & & 1 & 3 & -3 & 1 & \\ & & & 1 & 3 & -3 & 1 \end{bmatrix} \frac{1}{\Delta d} \in \mathbb{R}^{(m-3)\times n} \tag{7}$$

For pixels on the edges, the second-order and first-order difference operators can be used. The regularization parameter is analog to the weight parameter for the prior equations in the LTD algorithm.

The regularization parameter determines the balance between data fidelity and regularization terms. Determination of optimum regularization parameter is an important step of the regularization method. However, the regularization parameter is problem and data dependent. There is no general-purpose parameter-choice algorithm which will always produce a good parameter. For simplicity, we use the method based on discrepancy principle (Hamarik et al., 2012). The regularization parameter $\mu$ is chosen from a finite section of a monotonic sequence. For each value of $\mu$, an optimal solution is derived by solving the inverse

problem. Then the discrepancy can be calculated. The regularization parameter is determined to be the highest value that makes

80 the discrepancy $\|\boldsymbol{L}\boldsymbol{c} - \boldsymbol{b}\|_2^2$ equal to $N_b\sigma^2$, where $\sigma$ is the standard deviation of the noise. In this study, the reconstructions varied only slowly with the regularization parameters. Therefore, precise selection of the parameter was not necessary. For computational efficiency, the regularization parameter was selected from four widely varying values. The one produced the smallest discrepancy was used.

**2.3 Variational interpolation and minimum curvature algorithm**

85 Splines are special types of piecewise polynomials, which have proved to be very useful in numerical analysis and have founded in many applications in science and engineering problems. They match given values at some points (called knots) and have continuous derivatives up to some order at these points (Champion et al., 2000). Spline interpolation is preferred over polynomial interpolation by fitting low-degree polynomials between each of the pairs of the data points instead of fitting a single high-degree polynomial. Normally, the spline functions can be found by solving a system of linear equations with

90 unknown coefficients of the low-degree polynomials defined by the given boundary conditions.

The variational approach gives a new way to find the interpolating splines and opens up directions for theoretical developments and new applications (Champion et al., 2000). The variational interpolation was motivated by the minimum curvature property of natural cubic splines, i.e., the interpolated surface minimizes an energy functional which corresponds to a physical bending energy. This principle provides a lot of flexibility in controlling the behavior of the generated spline. Given an observation $z_k$

95 ($k$=1, …, $N$) measured at $k$-th point whose position vector is $\boldsymbol{r}_k$, a spline function $F(\boldsymbol{r})$ interpolating the data points can be found through variational approach by minimizing the sum of the deviation from the measured points and the smoothness seminorm of the spline function

$$\min_{F} \ \sum_{k=1}^{N} |F(\boldsymbol{r}_k) - z_k|^2 + \ \omega I(F) \tag{8}$$

where $\omega$ is a positive weight, $I(F)$ denotes the smoothness seminorm. The seminorm can be defined in various forms. The

100 commonly used ones are the first, second, third derivatives or their combinations. The solution of the minimizing problem is spline functions, which can also be found by solving a Euler-Lagrange differential equation corresponding to the given seminorm (Briggs, 1974).

We can see that the minimizing problem of Eq. (8) has a similar form to the Tikhonov regularization, but with a more flexible regularization term. The problem is that the variational interpolation is based on given data points, while the tomographic

105 reconstruction is based on measured line integrals. However, we show in this study that the variational approach for interpolation can also be applied to the latter problem to produce a smoothness solution having similar effect of spline interpolation. Also, based on different seminorms, we can formulate many different reconstruction algorithms. In this way, a new minimum curvature (MC) algorithm was proposed in this study.

Assuming the unknown concentration distribution is described by a function $f(x,y)$, $(x_k, y_l)$ is the lowest coordinates of the $j$-th

110 pixel at row $k$ and column $l$ of the 2-D grids, then the concentration $c_j$ equals to the average concentration of the pixel

$$c_j = \frac{1}{(\Delta d)^2} \int_{x_k}^{x_{k+1}} \int_{y_l}^{y_{l+1}} f(x,y)dxdy \qquad (9)$$

The minimization problem according to the variational approach is formulated as

$$\min_f \ \sum_{i=1}^{N_b} \sum_{j=1}^{N_c} \| L_{ij}c_j - b_i \|_2^2 + \ \omega I(f) \qquad (10)$$

For MC algorithm, we define seminorm according to the minimum curvature approach, which is used in the geographic data interpolation to seek a 2-D surface having continuous second derivatives and minimal total squared curvature (Briggs, 1974). The minimum-curvature surface has an analogy in elastic plate flexure and approximates the shape adopted by a thin plate flexed to pass through the observation data points with a minimum amount of bending. This method generates the smoothest possible surface while attempting to honor the observation data as closely as possible. The seminorm in the MC algorithm is defined to be equal the total square curvature

$$I(f) \ = \int \int \left( \frac{\partial^2 f}{\partial x^2} + \frac{\partial^2 f}{\partial y^2} \right)^2 dxdy \qquad (11)$$

This integral needs to be discretized according to the grid division. The discrete total square curvature is

$$I = \sum_{k=1}^{n} \sum_{l=1}^{m} (I_{k,l})^2 \qquad (12)$$

where $I_{k,l}$ is the curvature at the $(k,l)$ pixel, which is a function of $C_{k,l}$ and its neighboring pixel values. In two dimensions the approximation to the curvature is

$$I_{k,l} = (C_{k+1,l} + C_{k-1,l} + C_{k,l+1} + C_{k,l-1} - 4C_{k,l})/(\Delta d)^2 \qquad (13)$$

To minimize the total squared curvature, we need

$$\frac{\partial I}{\partial C_{k,l}} = 0 \qquad (14)$$

Combining Eq. (11), (12), and (13), we get the following difference equation

$$\begin{aligned} &[C_{k+2,l} + C_{k,l+2} + C_{k-2,l} + C_{k,l-2} \\ &+ 2(C_{k+1,l+1} + C_{k-1,l+1} + C_{k+1,l-1} + C_{k-1,l-1}) \\ &- 8(C_{k+1,l} + C_{k-1,l} + C_{k,l-1} + C_{k,l+1}) + 20C_{k,l}]/(\Delta d)^2 \ = 0 \end{aligned} \qquad (15)$$

This equation is appended at each pixel as a smoothness regularization. Therefore, there is only one prior equation at each grid instead of two equations in the LTD algorithm. For pixels on the edges, we set the approximation of the first and second derivatives to be zeros. Assuming $M$ is the kernel matrix of the prior equations, the reconstruction is to minimize the following problem

$$\min_c \ \| Lc - b \|_2^2 + \lambda \| Mc \|_2^2, \text{ subject to } c \geq 0 \qquad (16)$$

where the parameter $\lambda$ is determined in the same way as determining the regularization parameter in the Tikhonov regularization method. Similar to the LTD approach, the resulted constrained system of linear equations is also over-determined and is solved by the NNLS optimization algorithm.

[Figure]

**Figure 2.** The Beam geometry and a 30 × 30 grid division of the site.

140    For conventional pixel-based reconstruction algorithms, the number of pixels (unknowns) should be no more than the number of beams (equations) in order to get a well-posed problem. Because only tens of beams are usually used in the ORS-CT applications, the resulted spatial resolution is very coarse. The GT algorithm is one way to increase the resolution. But it needs several steps to complete the whole translation because each translation uses a different grid division and the reconstruction process needs to be conducted for each grid division. In the MC algorithm, we use only one division of high-resolution grids

145    directly during the reconstruction. The resulted system of linear equations is still determined because of the smoothness restriction at each pixel. As shown in Fig. 2, 30 × 30 pixels are used in the MC algorithm instead of the 6 × 6 pixels in the NNLS approach. Under this configuration, the number of linear equations for the LTD algorithms is approximately 38+30×30×2=1838. While the number for the MC algorithm is about 38+30×30 = 938. We can see that the MC approach reduced the number of linear equations by approximately half comparing to the LTD algorithm. The smoothness seminorm of

150    the MC algorithm will guarantee a smooth solution. This smooth effect is similar to the spline interpolation applied after the reconstruction process, except that it is achieved during the inverse process. This is important because an interpolation after the reconstruction cannot correct the error resulting from the reconstruction based on coarse spatial resolution. While the MC approach evaluates the discrepancy based on the high-resolution values that are the same as the reconstruction outcomes. Errors due to coarse spatial resolution are corrected during the process.

155    …

**2.5 Evaluation of reconstruction quality**

…

In this paper, a measure using the resolution matrix is also used to predicate the reconstruction error due to different regularizations approaches. Resolution matrices are commonly used to determine whether model parameters can be

160 independently predicted or resolved and how regularization limits reconstruction accuracy (Twynstra and Daun, 2012; von Clarmann et al., 2009). Ignoring the non-negative constrains, the generalized inverse matrices for the NNLS, LTD, and MC algorithms can be found by

$$G_{NNLS} = (L^T L)^{-1} L^T$$

$$G_{LTD} = (L^T L - \mu^2 D_3^T D_3)^{-1} L^T$$

165 $$G_{MC} = (L^T L - \lambda^2 M^T M)^{-1} L^T \tag{21}$$

The resolution matrix is defined as $R = GL$. The reconstruction error is given by

$$\delta c = c_{model} - c_{exact} = (R - I)c_{exact} - G\delta b \tag{22}$$

where $c_{model}$, $c_{exact}$ is the model-predicated and the exact concentrations respectively, $\delta b$ is the perturbation of the observation, $I$ is the identity matrix, $(R - I)c_{exact}$ is the regularization error caused by the inconsistency between the measurement data

170 equations and the prior information equations, $G\delta b$ is the perturbation error.

For the LTD and MC approaches using high-resolution grids, the kernel matrix $L$ is rank-deficient, and the regularized solution is robust to perturbation error over a wide range of regularization parameter. Thus, the perturbation error is negligible, and the reconstruction error is dominated by regularization error (Twynstra and Daun, 2012). Because the resolution matrix is only determined by the beam configuration and the regularization approach, it is independent of the actual concentration

175 distribution. Due to this reason, it is better to be used to evaluate different beam configurations which have considerable influence on the reconstruction accuracy. However, in this study the beam configurations are fixed. We can therefore use the resolution matrix to measure different regularization approaches. In an ideal experiment, $R = I$, which implies that each unknown pixel value can be independently resolved from the measurement data. The regularization term forces the off-diagonal terms in $R$ to be nonzero, making the estimated concentration of each pixel a weighted average of the concentration

180 of the surrounding pixels. We can use the Frobenius distance between $R$ and $I$ defining a measure of fitness to predicate the reconstruction error (Twynstra and Daun, 2012).

$$\varepsilon = \frac{1}{N_c} \|R - I\|_F^2 \tag{23}$$

**3 Results and discussions**

…

185 ### 3.5 Fitness

The contour plots of the resolution matrices for the LTD and MC algorithms are shown in Fig. 4. (a) and (b). The fitness values are 1.4411 and 1.3878 for the LTD and MC algorithm respectively. The MC algorithm shows slightly better performance. The off-diagonal elements are not zeros. The reconstructed concentration at each pixel is a weighted average of the concentrations of the surrounding pixels because of the smoothness regularization. Each row of the resolution matrix can be regarded as

smoothing weights. As the pixels have a 2-D arrangement, we show the 2-D display of the row of the 106th pixel (row and column indices are 4,16 in 2-D pixels) in the resolution matrix for the LTD and MC algorithms in Fig. 4. (c) and (d) as an example, from which we clearly see the dependence on the beam geometry. In this study, the beam configuration is fixed, thus 
[revised manuscript text omitted]

---

## Author Response (AR1)

**Response to comments of reviewer 1**

We thank the reviewer for the helpful comments and suggestions, and for careful reading of the manuscript. Listed below are our itemized responses, with the original comment/question displayed in italics.

**General comments**

*The paper faces the problem to introduce the "smoothness" a priori information in the tomographic reconstruction of atmospheric chemicals based on optical remote sensing. In particular, a new minimum curvature (MC) algorithm is proposed and applied to multiple test maps. The performance of the new algorithm is compared with that of other existing algorithms. The MC algorithm shows almost the same performance as the low third derivative (LTD) algorithm but with*
*significantly less computation time.*

*I think that the subject is correctly presented in the introduction and sufficiently put in the context of the existing literature on the argument; instead, I find that the description of the method is not given in all needed details. I suggest to improve the description of the method and below I give some suggestions.*

*I think that the paper deserves the publication on AMT after that the following issues are considered.*

Thank you for this comment. We have improved the description according to your suggestions. Please see the responses below for specific changes we have made to the manuscript.

**Specific comments:**

(1) *In the Tikhonov approach, an important issue is the choice of the value that is given to the regularization parameter,*
*because this value determines how much a priori information goes into the results. In the paper, it is specified only "the regularization parameter is set to be inversely proportional to the grid length". I suggest describing the criterion that it has been followed for the choice of this parameter.*

This is a good suggestion. A weight needs to be assigned to each equation depending on the uncertainty of the observation. Assuming the analyzers have the same performance, the uncertainty is mainly related to the path length. Therefore, equations
are assigned weights inversely proportional to path length to make sure different paths have equal influences. In this study, the lengths of the laser paths are approximately equal to each other. So their weights were all set to be the same value which is scaled to be 1. The weights for the third-derivative prior equations were assigned to be the same value of $w$ because they were all based on the same grid lengths. w was determined in the same way of determining the regularization parameter $\mu$ in the Tikhonov regularization

The regularization parameter determines the balance between data fidelity and regularization terms. Determination of optimum regularization parameter is an important step of the regularization method. But the regularization parameter is problem and data dependent. There is no general-purpose parameter-choice algorithm which will always produce a good parameter. For simplicity, we use the method based on discrepancy principle. The regularization parameter is chosen from a finite section of a monotonic sequence. For each value of $\mu$, an optimal solution is derived by solving the inverse problem.
Then the discrepancy can be calculated. The regularization parameter is determined to be the highest value that makes the discrepancy equal to $N_b\sigma^2$, where $N_b$ is the number of beams, $\sigma$ is the standard deviation of the noise. In this study, the reconstructions varied only slowly with the regularization parameters. Therefore, precise selection of the parameter was not necessary. The regularization parameter was selected from four widely varying values. The one produced the smallest discrepancy was used.

We have added a separate paragraph at the end of section 2.3 to discuss how the weights and regularization parameter are determined.

*(2)  It would be interesting to know if the algorithm is able also to produce a diagnostics of the results. Generally, a procedure that solves an inverse problem provides also an estimation of the errors (more in general of the covariance*
*matrix) of the products. Furthermore, it would be useful to have quantities (such as the averaging kernel matrices obtained in the case of retrieval of atmospheric vertical profiles) that provide the sensitivity of the result to the true state, which are useful also to estimate the spatial resolution of the result.*

This is a very good idea. The averaging kernel matrix is not much meaningful for inversion without prior information. But it can be used for the regularized problem. In the 2-D tomographic reconstruction, the averaging kernel is considerably affected
by the beam geometry and is better to be used as a measure to evaluate the beam configuration. But it also reflects the regularization error given the same beam geometry in this study.

A new measure based on the averaging kernel matrix have been added to the manuscript to determine whether the concentration can be independently predicted and how regularization limits reconstruction accuracy. The new fitness measure is defined as the average Frobenius distance between the resolution matrix and the identity matrix to predict the
reconstruction error. The MC algorithm shows slightly better performance than the LTD algorithm with a value of 1.3878 comparing to 1.4411. The off-diagonal elements are not zeros. The reconstructed concentration at each pixel is a weighted average of the concentration of the surrounding pixels because of the smoothness regularization. Each row of the resolution matrix can be regarded as smoothing weights. The 2-D display of the row of the averaging matrix gives a clear dependence of the beam configuration.

Section 2.5 and 3.5 have been revised for detailed description.

*(3)  Line 43: I suggest to put a reference for the Radon transformation.*

The reference has been added.

*(4)  Line 141-149: In the description of the LTD algorithm it is not clear which are the equations of the system that has to be solved. I understand that for each cell we have two equations obtained setting to zero the third derivatives (in both direction x and y, I suppose, but it is not specified). Then, which are the other equations? Those obtained to look for the minimum of Eq. (3)? Please explain in detail which are the equations of the system that has to be solved.*

At each grid pixel, two additional third-derivative priori equations are set up by setting the approximations of the third-order
derivatives to zero. This results in a new system of linear equations $Tc=0$ for all the pixels, where $T$ is the prior kernel matrix. These prior equations are appended to the original system of linear equations defined by the PIC observations, resulting a new augmented system of linear equations. Assuming the new augmented kernel matrix is $A$, observation vector is $p$, then the new system of linear equations is **Ac=p**. The inverse problem is to solve the new system of linear equations with non-negative constraints.

We have rewritten the contents describing LTD algorithm in Sec. 2.2.

*(5) Line 159-160: From Eq. (7) I understand that the seminorm is a number relative to the whole field, therefore, I do not understand the meaning that "the seminorm can be calculated at each pixel". Then, which is the summation mentioned in the text? I think that a more clear explanation is needed.*

Thank you for pointing this out. It is correct that the seminorm, which is the total squared curvature, is defined on the whole field. The total square curvature is the sum of the squared curvature at each grid pixel. To minimize the seminorm, we need its partial derivative with respect to the concentration at each grid pixel to be equal to zero. Thus, we get a difference equation at each grid, which is appended into the original linear equation to form a new system of linear equations.

A revised description including the derivation process has been added to section 2.3.

**Technical corrections:**

*(1) The authors introduce many acronyms, but not all of them are then used. I suggest introducing only the acronyms that are used several times in the paper.*

Thanks for this good suggestion. We have checked the acronyms and removed those used only once.

*(2) Line 26: equality ---> quality*

Corrected.

*(3) Line 85: necessary ---> need*

Corrected.

*(4) Line 136: what is the superscript 21 after "problem"?*

The number has been replaced with a reference.

*(5) Line: 174: well-posted ---> well-posed.*

Corrected.

*(6) Line 212: It ---> it*

Corrected.

*(7) Line 242: increase ---> increases*

Corrected.

*(8) Line 286: equality ---> quality*

Corrected.

**Response to comments of reviewer 2**

We thank the reviewer for the helpful comments and suggestions, and for careful reading of the manuscript. Listed below are our responses, with the original comment/question displayed in italics.

**GENERAL COMMENTS**

====

*The paper describes a minimum curvature based regularization schemed for deriving 2-D trace gas concentrations from optical remote sensing and tomography. The chosen regularization scheme is sensible and the method seems to be an improvement over the state of the art in the field The topic fits the journal.*

*The textual description is severely lacking and a rewrite to better guide the reader through the numerous methods is necessary. The description of the compared methods is severly lacking mathematical rigour and precise definitions causing*

*the research to be not replicable in its current state.*

*I believe that the paper can only be published after a major revision and restructuring. See below for some general guidelines and a number of specific issues.*

Thank you for the comments. We have done a major revision following your suggestions and addressed the issues. Please see the responses below for specific changes we have made to the manuscript.

**MAJOR COMMENTS**

======

**Precise **Mathematical** description**
* * *
*The problem requires a much more precise mathematical introduction with clear definitions of employed terms. The paper gives a wide overview over several methods from literature and introduces many of these and related terms without clear definitions. At least those discussed later should be introduced well enough to follow the paper without further referencing.*

*The continuous and discrete view of the problem needs to be separated and the relationship clarified (see specific comments). Very often it is useful to specify the formulas for the continuous case and then "simply" discretize the resulting*

*integrals and derivatives. In this case one achieves results that are*

*less dependent on the chosen discretization/gridding. This is particularly true as the sampling distance is (wrongly) used in the regularization strength instead of the integrals itself.*

Thanks for the suggestions. We have redefined the symbols in this manuscript. Clear and precise descriptions are given. The continuous and discrete views have been distinguished by using different symbols. The issue with the sampling distance has been corrected. A new section of 3.6 has been added to discuss the influences of the grid size.

**Motivation for minimum curvature**
* * *
*This chosen regularization is criticized for producing oversmoothened results. The major question is, what kind of regularization term would best describe the a priori information. The Laplacian is an obvious choice due to its relationship with the Poisson-Equation. If diffusion is the major process than a norm related to an exponential covariance would be very useful (see "Inverse Problem Theory" by Tarantola); also here, the Laplacian pops up at least for the 3-D exponential covariance. It would be interesting to motivate the choice of regularization form the underlying physics.*

*There is also a host of literature with respect to regularization for optical remote sensing methods from nadir and limb sounding satellites. It would be very interesting to put this method into this context and/or discuss the statistical angle.*

It is true that the prior information should be based on the underlying physical process. The smoothness *a priori* information is most commonly used and very suitable for the diffusion process. More realistic prior information may be better for specific applications. For example, we have almost done investigating more complicated regularization based on the information derived from the air dispersion models for the diffusion-convention process. But the smoothness is the basic feature for all the problems in ORS-CT applications. In this study, we focus on better understanding the characteristics of this smoothness regularization. This is done by adopting the variational interpolation technique into the tomographic reconstruction process. Therefore, the smooth effect is closely related to the spline functions which has been well studied in the literature. Improving the performance of algorithms by using different seminorm is possible.

The techniques used in similar field should be very inspiring. We are also expecting to explore more ideas from them. We also realize that currently the available algorithms for the 2-D tomographic mapping of the air contaminants are still very limited. This may be caused by the fact that the inversion problems are not exactly the same as those in similar area, especially when we look at the beam configurations and underlying distributions which considerably affect the performance of the algorithms. Therefore, algorithms need to be modified or adopted to be used in the specific applications. Overall, there are still lots of work to do in this area to promote the practical application of the ORS-CT technique.

**Diagnosis**
* * *
*Due to the choice of a grid with more unknowns than measurements, diagnostics become more important. This can be done in a simple fashion with "resolution" measures. Rodgers' "Inverse Methods for Atmospheric Sounding" shows in great detail*

*what kind of diagnostic quantities are relevant (resolution, measurement contribution, smoothing error, uncertainties...)*

The nearness, peak location error, and the exposure error are the commonly used measures to evaluate the image quality. They provide direct measure of the performance only based on the final reconstruction results and independent of the reconstruction approaches. So they do not reveal the internal feature of the algorithms. In this view, the averaging kernel matrix is a good complementation.

According to the suggestion, we find that the averaging kernel matrix (resolution matrix) is commonly used in the atmospheric sounding but seldomly in the 2-D tomographic reconstructions. In the latter case, it is largely affected by the beam geometry and is better to be used as a measure to evaluate the beam configuration. But it also reflects the regularization error given the same beam geometry in this study.

A new measure based on the resolution matrix have been added to the manuscript to determine whether the concentration can be independently predicted and how regularization limits reconstruction accuracy. The new fitness measure is defined as the average Frobenius distance between the resolution matrix and the identity matrix to predict the reconstruction error. For the result, the MC algorithm shows slightly better performance than the LTD algorithm with a fitness value of 1.3878 comparing to 1.4411. The off-diagonal elements in the resolution matrix are not zeros. The reconstructed concentration at each pixel is a weighted average of the concentration of the surrounding pixels because of the smoothness regularization.

Each row of the resolution matrix can be regarded as smoothing weights. The 2-D display of the row of the averaging matrix gives a clear dependence of the beam configuration.

Section 2.5 and 3.5 have been revised for detailed description.

**SPECIFIC COMMENTS**

=====

**line 35**
* * *
*"which can detect a large area in situ and provide near real-time information" Maybe cover? Also "in situ" may be an unconventional use for a remote sensing instrument, depending on the community.*

We have updated the description as following: "The ORS-CT technique provides a powerful tool for sensitive mapping of air contaminants over a kilometer-size area in real time."

**line 43**
* * *
*To be precise both an infinite number of beams and beams of infinite length are required. One typically assumes a zero signal outside the reconstruction domain, which, for your problem, is a very reasonable assumption and at least alleviates the latter condition. How does the finite number of beams affect the solution?*

It is a good idea to apply the zero-signal assumption outside of the domain, but in some applications of monitoring the air pollutants, this assumption may not be true. This is usually the case for mapping the concentration distribution over a
horizontal plane because the plume may extend for a long distance outside of the mapping area, and the ORS-CT system cannot cover very large area due to the instrument limitation or deployment difficulties. We think this situation is different from the medical CT which reconstructs a slice with a clear boundary.

The number of beams affects the spatial resolution of the reconstructed map. The concentration of an area can be theoretically reconstructed if it is independently sampled by the beams. The beam number and geometry determine the size
of an area that can be sampled independently. In the view of the reconstruction algorithm, the number of beams determines the number of equations of the inverse problem. The unknows are the concentration values of the grid pixels. To make the system of equations to be not ill-posed (underdetermined), it is required that the number of pixels should be no more than the number of beams. Otherwise, the problem of indeterminacy will occur and artifacts may appear on the reconstructed map. Because only tens of beams are used in ORS-CT, the resulting spatial resolution produced by the traditional pixel-based
reconstruction algorithms is very coarse. This issue can be mitigated by adding the smoothness information to the inversion. We have rewritten the paragraph two in the introduction chapter to discuss the problem of the limited beam number.

**line 44**
* * *
*"Series-expansion-based methods". Unclear what is meant here. Cite needed. The explanation sounds like a simple discretization, transforming the "continuous problem" of finding a function over L2 to a discrete problem of identifying a number of samples.*

A reference has been added as suggested (Censor, 1983). The series expansion methods are distinguished from the transform methods by discretizing the problem at the very beginning whereas with transform methods the continuous problem is
handled until the very end when the final formulas are discretized for computational implementation. In practice, the reconstruction domain is divided into discrete grid pixels. The pixel concentrations result from the linear combination of finite number of basis functions defined on the domain. We have updated the description in paragraph two.

**line 48**

-------

*Also medical reconstruction techniques employ discrete samples and basis functions. With many more samples, obviously.*

The number of beams in the ORS-CT is very limited comparing to the medical CT (tens vs hundreds of thousands), which poses new challenges for the reconstruction techniques. The main issues are the coarse spatial resolution and the ill-posed inverse problem. For limited data tomographic reconstruction, the system of linear equations is rank-deficient. Additional *a*
*priori* information like smoothness must be used to improve the performance. We have revised the texts in paragraph two to emphasis these challenges.

**line 49**
* * *
*I do not understand the difference between a pixel based approach and a basis function based. A pixel is a basis function with rectangular, non-overlapping support.*

The reviewer's understanding is accurate. There is ambiguity related to the classification of the methods. A pixel is one of the simplest basis functions with unit value inside the pixel and zero outside. The "basis function" in the manuscript is actually a "smooth basis function". To eliminate this ambiguity, we use only the "series expansion method", which is based on a linear
combination of finite set of basis functions, as a category comparing to the transform method. And define the "pixel-based method" and "smooth basis function method" as two types of approaches using different basis functions. The text has been revised accordingly in paragraph two and three.

**line 51**

-------

*"best" in what sense?*

We have updated the description as following text: "The inverse problem is to find the optimal set of pixel concentrations by minimizing the error function constructed based on some criteria, including minimizing the $L_2$ norm of error (finding a least-squares solution), maximum likelihood (ML), maximum entropy, etc."

**line 52**
* * *
*Typically a basis is a basis of a (vector) space. Which space is spanned here? Is the full space spanned or only a subspace thereof?*

The two-dimensional reconstruction domain is divided into $N_c=m\times n$ grid pixels. For the pixel-based methods, the space is the $N$-dimensional real space which is fully spanned by the $N$ pixel-based basis functions. For smooth basis function methods, the space is the functions representing the distribution which may not be fully spanned depending on the choice of the basis functions. Taking SBFM algorithm as example, usually there are no more than five bivariate Gaussians are used so it can only represent some specific distributions.

**line 53**
* * *
*What parameters? Typically one derives pre-factors of normed basis functions.*

Pre-factors are one type of parameters. But there may be more other parameters which need to be determined. For example,
for the generalized Kaiser-Bessel window function, there are three parameters: sampling distance, window duration, and window shape. For the bivariate Gaussian in the SBFM algorithms, there are six parameters: normalizing coefficient, correlation coefficient, 2-D peak locations, and two standard deviations. But there are two ways dealing with these basis functions. First, the parameters are determined before the reconstruction according to mathematical analysis. Second, these parameters are kept unknown and the problem is to fit them to the observation data. The SBFM algorithm works in the latter way. We have revised the description of the SBFM algorithm to explain these parameters and how the algorithm solves the problem in paragraph three.

**line 54**
* * *
*What equations? Why are those equations non-linear? Typically this problem would be linear even for non-trivial basis functions.*

The equations are defined by the observed PIC data and the predicted values. The non-linear issue is specific for the SBFM algorithm, in which case the concentration values are calculated from linear combination of several bivariate Gaussians. But these Gaussians have unknown parameters as mentioned in the previous answer. The problem is to fit these parameters to the measurement data instead of getting the concentration values directly. Therefore, the bivariate Gaussians have to be calculated every time when a new set of parameters is proposed during the optimization process, which makes the problem non-linear and very computation intensive. We have revised the description of the SBFM algorithm to give more accurate explanation in paragraph three.

**line 55**
* * *
*Best in what sense?*

For the SBFM algorithm, the problem is to find the optimal set of parameters which best fits the observed PIC data by minimizing an error function using least square criterion. The text has been revised.

**line 56**
* * *
*Too general. Fits to nearly any problem. What error function based on which criteria?*

The least square criterion is to minimize the sum of the squared errors between the observed and model-predicted PICs. The maximum likelihood criterion is to maximize the probability of the PIC observations given the distribution of the errors. The maximum entropy criterion is to maximize the entropy of the reconstructed maps given the average concentration of the map is known. The manuscript has been updated accordingly.

**line 58**

-------

*Define ill-posed. Define very large (millions?).*

The ill-posed problem refers to solve the underdetermined system of linear equation. For traditional approaches like ART and NNLS, the number of equations is determined by the number of beams which is usually hundreds of thousands for medical CT. But for ORS-CT, the number is only about tens. So "very large" is not accurate and has been deleted.

**line 61**
* * *
*Many classes of non-linear problem can be solved efficiently by deterministic methods. Particularly convex optimization problems such as this.*

It is true that the non-linear problems can be solved by deterministic methods. We have revised the manuscript to make the description to be more accurate as following. First, the system of equations defined by the measured and predicted PICs are non-linear because of the presence of the bivariate Gaussians with unknown parameters. Second, the searching of the best-fit set parameters can be solved by iterative minimization procedure such as simplex method or simulated annealing. Third, simulated annealing method was used in the literature to find a global minimization when the cost function has many local
minima. This minimizing process was computation intensive. It is possible to improve the speed by modifying the simulated annealing algorithm or using other optimization techniques. But currently, it is still very slow comparing to solving a system of linear equations.

**line 65**

-------

*Exploiting previous (a priori) knowledge of a problem is almost always key in inverse problems. Doesn't dispersion/diffusion suggest a Laplacian as regularizing term? Is there a physical relationship between the dispersion processes and the minimum curvature?*

Yes. The Laplacian arise in the diffusion equation. Under steady state, the diffusion process is described by the Laplace's
equation. In general case, the dispersion process is described by the convection-diffusion equation with additional convection and source/sink terms comparing to the diffusion process. The minimum curvature principle seeks a two-dimensional surface which minimizes the total squared curvature (the Laplacian power). The minimizing problem is also equivalent to solving a biharmonic equation. Therefore, we can investigate the similarity and difference between different algorithms from the equivalent equations. We have added more physical interpretation of the minimum curvature method in
2.3 section.

**line 69**
* * *
*This is the first time NNLS is mentioned in the main text and a cite should be placed here with more detail. The EPA cite does not detail the NNLS algorithm.*

A reference has been added (Lawson and Janson, 1995).

**line 74**
* * *
*If the number of unknowns is smaller than the number of measurements, such a problem may still be solved by using pseudo-inverses and or regularization techniques, which are computationally cheap.*

There are several things to mention here. It is true that an underdetermined linear system can be solved by pseudo-inverse method, which selects the solution with minimum Euclidean norm in this case. However, there is a problem of overfitting. This solution may not be physically realistic. It does not have the smoothness feature. Another thing is that the reconstruction is a constrained problem (non-negative solutions). The pseudo inverse may not generate a solution satisfying the constraints. Overall, the reconstruction of the underdetermined linear system is an ill-posed problem. Regularization techniques are needed to find an appropriate solution by adding additional realistic *a priori* information to the problem. For the ORS-CT reconstruction problem, the smoothness *a priori* information has been proven to be a good choice.

**line 88**
* * *
*"But the theory basis of the LTD algorithm was not clearly given". By whom? This paper is so far not helping in this regard.*

The LTD algorithm applies third-order derivative constraints to the solutions. But the underlying theory of this operation was not clearly defined in the literature. This prevents us from studying the characteristics of these constraints and getting a broad picture of the method. This manuscript first identifies the LTD method as one special case of the well-established Tikhonov regularization. Then it studies more flexible smoothness constraints through the theory of variational interpolation, based on which the characteristic of different smoothness constrains are well understood through the close connection with the spline functions. Finally, we successfully adopted the variational approach from spatial interpolation problem into the tomographic reconstruction problem. We have rewritten the last two paragraphs in the introduction section to show these connections, and updated section 2.2 and 2.3 to give more description of the theories.

**line 92**
* * *
*What is regularization?*

For the inverse or optimization problem, regularization is the process of adding information in order to solve an ill-posed problem or to prevent overfitting. The regularization term (penalty) imposes a cost on the optimization function to make the optimal solution unique.

**line 95**

-------

*Interpolation theory typically deals within interpolation of (mostly discrete) data. How does that relate to the problem at hand?*

This is one point that we want to show in this study. The interpolation techniques are based the given sample points, which is different from the tomographic reconstruction where only the line integrals are known. But we find that the variational
interpolation technique can be adopted into the reconstruction process to produce a smooth solution. The connection is established based on two facts: (1) variational method is another way achieving the spline interpolation because the interpolating splines can be derived as the solution of certain variational problems of minimizing a seminorm defined by an integral consisting of different derivatives or their combinations; (2) this smoothness seminorm can also be used as a smoothness regularization factor for the tomographic reconstruction problem. In this way, the smooth effect similar to the
spline interpolation can be achieved during the reconstruction process. We also find that other interpolation techniques can also be adopted in a similar way. We have rewritten the last paragraph of the introduction and section 2.3 to describe this idea.

**line 98**

-------

*"The solution to this problem is a set of spline functions." Please be precise about this. The algorithm derives, necessarily a vector. This vector can be interpreted in a various of ways. Of particular import is how it is interpreted by the "forward model", because that determines what is fit. This interpretation may differ from the interpretation for the regularisation term, but this introduces necessarily an error. One should be clear about that. Typically one sees the regularization term as an*
*approximation: as the computation of derivatives by finite differences is inherently approximate. In the case that the gridding is very fine, the approximation error becomes small, and the point is moot, but this discussion is missing here. The discretization error has not been discussed and thus cannot be neglected. It is sensible to represent the 2-D field to be reconstructed here as a 2-D spline both in the forward model and the regularization term. This would remove approximation errors at the cost of a more complicated algorithms. Either way, the distinction and used assumptions must be made explicit*
*and errors discussed.*

Thanks for the good suggestions. The most important fact is that the variational approach is another way to find the interpolating splines. Therefore, by solving the minimization problem with the smoothness seminorm penalty, we will get a smooth solution similar to the effect of applying spline interpolation in the view of the forward model. Normally, interpolation is based on the given data points, this study, however, successfully applied it to the problem based on line
integral data. The final solution is smooth and also has characteristics related to the spline interpolation. But the "interpolation" is achieved during the inversion process instead of after it, which is a key to improve the reconstruction accuracy. We have rewritten the last two paragraphs in the introduction section to clarify this relationship and the idea. Section 2.3 have also been revised for the description of the relationship between the variational interpolation and spline function.

A new section of 3.6 has been added to discuss the discretization error due to the use of different grid sizes. The changes of the nearness, peak location error, exposure error, and computation time with respect to the pixel number were recorded using different grid divisions. The results show that the nearness, peak location error, and exposure error generally illustrate decreasing trends when increasing the pixel number. The MC algorithm shows slightly better performance than the LTD algorithm with the increase the pixel number. The performance improvements become slow for both algorithms when the division is finer than 24 × 24. On the other hand, the computation time shows approximately exponential growth trend with the increase of the pixel number. But the LTD algorithm has a faster increasing rate than the MC algorithm. So there should be a balance between the performance and the computation time.

**line 102**
* * *
*Maybe a bit more of the theory should be described to make this more obvious to the reader.*

The Tikhonov regularization is a well-known technique to solve the ill-posed inverse problem. The Tikhonov $L_2$ regularization uses a penalty term defined by the squared norm of the $i$th-order derivative of the function to produce a smoothing effect on the solution. The variational interpolation is motivated by the minimum curvature property of natural cubic splines. It is another way achieving the spline interpolation by the fact that the interpolating polynomial splines can be derived as the solution of certain variational problems of minimizing an integral consisting of different order derivatives or their combinations. The interpolating data points can be found through minimizing the sum of the deviation from the measured points and the smoothness seminorm. We have updated the description in the last two paragraphs of the introduction and section 2.2 and 2.3 to give more explanation of the Tikhonov regularization and the variational interpolation theories.

**line 104**
* * *
*Please properly and mathematically introduce the corresponding biharmonic equation and the smoothness seminorm.*

The seminorm is defined as the total squared curvature of the concentrations in the MC algorithm. The minimizing problem is equivalent to solving a corresponding biharmonic equation. The definition of the biharmonic equation has been removed because we did not use it for finding the solution in this study. The total squared curvature is first discretized. Then in order to minimize the seminorm, we need its partial differential with respect to the pixel concentration to zero, which leads to a difference equation at each pixel. This equation is appended at each pixel as a smoothness constraint. Thus, a new system of liner equations is set up. We have rewritten section 2.3 to mathematically define the smoothness seminorm.

**line 106**
* * *
*Why does it half the number of equations?*

In the LTD algorithm, two equations are added at each pixel defined by the third-derivative prior information. While in the MC algorithm, only one equation is added that is derived from the minimizing of the total squared curvature. if we use 38 beams and a 30 x 30 grid division, there will be 38+30×30×2=1838 equations for LTD, and 38+30×30=938 equations for

MC. So the MC approach reduces the number of equations by approximately half. We have added the explanation in section 2.3.

**line 107**
* * *
*If the number of grid points increases what does that mean for the amount of information contained in the results and to what degree are the resulting "pixels" correlated? I.e. how well is the result resolved?*

We have added a new section 3.6 to show the influence of different grid sizes. The changes of the nearness, peak location error, exposure error, and computation time with respect to the pixel number were recorded using different grid divisions. The results show that the nearness, peak location error, and exposure error generally illustrate decreasing trends when increasing the pixel number. The MC algorithm shows slightly better performance than the LTD algorithm with the increase of the pixel number. The performance improvements become slow for both algorithms when the division is finer than $24 \times 24$. On the other hand, the computation time shows approximately exponential growth trends with the increase of the pixel number. But the LTD algorithm has a faster increasing rate than the MC algorithm.

To conclude, the reconstruction performance is improved for both LTD and MC algorithms with the increase of the pixel numbers, but at the cost of exponential growth of the computation time. And the improvement become small when the resolution is higher than a certain threshold value ($24\times24$ in this study). So there should be a balance between the performance and the computation time.

Please refer to section 3.6 for the detailed description.

**line 137**
* * *
*A very important interpretation of this approach is the statistical one (optimal estimation), where R^TR can be interpreted as precision matrix codifying a priori information about the given distribution (i.e. smoothness). Please discuss.*

Thanks for the suggestion. A measure of fitness based on resolution matrix (averaging kernel matrix) has been used to investigate the regularization error caused by the inconsistency between the PIC equations and the prior information equations. This measure was defined as the average Frobenius distance between the resolution matrix and the identity matrix to predict the reconstruction error. Section 2.5 was added to define the measure.

A new section of 3.5 has been added to evaluate the results based on the fitness measure. The MC algorithm shows slightly better performance than the LTD algorithm with a fitness value of 1.3878 comparing to 1.4411. The off-diagonal elements are not zeros. The reconstructed concentration at each pixel is a weighted average of the concentration of the surrounding pixels because of the smoothness regularization. Each row of the resolution matrix can be regarded as smoothing weights. The 2-D display of the row of the averaging matrix gives a clear dependence of the beam configuration, while the diagonal elements may not provide much information in this case.

**line 139**
* * *
*In what sense is this an approximation? The given formula is discrete already, as such R_i is not a derivative, but a finite difference operator.*

The approximation means the operator is not the actual derivative operator but an approximation of the derivative operator. We have changed them to "finite difference operator".

**line 142**
* * *
*The nomenclature is highly unusual. Typically derivatives are defined for continuous functions. c was so far a vector. This is an approximation of the third-order derivative of a function in y-direction by finite differences. And even then, the division by the grid-distance is missing. In this form, the regularisation is grid-dependent and would change in strength for different* 500 *grid sizes, which requires a re-tuning of regularisation strength for every change of grid size. Please take the grid size into account.*

Thanks for the correction. We have revised the symbol definitions in this manuscript. The discrete pixel concentrations can be arranged in both one-dimensional (1-D) and two-dimensional (2-D) forms. $c$ is defined as a vector containing the concentrations in 1-D form with index of $j$. $C$ is now a matrix containing the concentrations in 2-D form with row and 505 column indices of $k$, $l$. The continuous distribution is described by a function of $f(x,y)$. The division of the grid-distance has been moved to the finite difference equations. Therefore, the regularization parameter is independent of the grid size. Section 2.2 and 2.3 have been revised for the definitions.

**line 148**

--------

*Why is the regularization parameter set to 1 over grid length? To compensate for the missing factor in R_3, the power three is missing. There is a host of literature discussing optimal choice of this parameter (L-curve, optimal estimation, etc.). Practically, it is a tuning parameter which often requires manual adjustments unless both measurements and a priori are very well understood.*

A weight needs to be assigned to each equation depending on the uncertainty of the observation. Assuming the analyzers have the same performance, the uncertainty is mainly related to the path length. Therefore, equations are assigned weights inversely proportional to path length to make sure different paths have equal influences. In this study, the lengths of the laser paths are approximately equal to each other. So their weights were all set to be the same value which is scaled to be 1. The weights for the third-derivative prior equations were assigned to be the same value of $w$ because they were all based on the 520 same grid length. The weight is analog to the regularization parameter in the Tikhonov regularization and determined in the same way.

The regularization parameter determines the balance between data fidelity and regularization terms. Determination of optimum regularization parameter is an important step of the regularization method. However, the regularization parameter is problem and data dependent. There is no general-purpose parameter-choice algorithm which will always produce a good parameter. For simplicity, we use the commonly used method based on discrepancy principle. The regularization parameter is chosen from a finite section of a monotonic sequence. For each value of $\mu$, an optimal solution is derived by solving the inverse problem. Then the discrepancy can be calculated. The regularization parameter is determined to be the highest value that makes the discrepancy equal to $N_b\sigma^2$, where $N_b$ is the number of beams, $\sigma$ is the standard deviation of the noise. In this study, the reconstructions varied only slowly with the regularization parameters. Therefore, precise selection of the parameter was not necessary. The regularization parameter was selected from four widely varying values. The one produced the smallest discrepancy was used.

We have added a separate paragraph to discuss how the weights and regularization parameter are determined at the end of section 2.3.

**line 148**
* * *
*What is meant by "setting the derivative to zero"? Formula (3) minimizes the expression and thus allows for non-zero derivatives unless the factor \mu is chosen to be very large.*

The derivatives are set to zero to generate constraint equations. That is how the third-derivative prior equations are set up
and added to the original equations defined by the observations. That is true that the final optimal solutions may not have zero derivatives everywhere. They are approximately zero.

**line 152**
* * *
*|c| is typically the absolute value of c. To describe more complicate regularization terms, one often uses a more general function Phi(c) mapping R^n to R^+ or a norm with a subscription like ||c||_\phi^2. Are you refering to Sobolev-Norms?*

Thanks for the suggestion. After revising the symbol definitions, *c* is a vector representing the discrete pixel concentrations. The seminorm is described by *I(f)* defined on the continuous distribution function of *f*. The discretization from of the seminorm is also provided in section 2.3 in the implement.

**line 155**
* * *
*The solution to the problem is a discrete vector c, whereby each element of c defines the concentration in one pixel (see (1)). A spline is something very different, as it is a continuous. Your problem is set up to be non-continuous by definition. Please*
*specify your model precisely and be careful with the distinction between the continuous and discrete view.*

Thanks for the correction. We have distinguished the continuous and discrete view by using different symbols. Now this minimizing problem is defined in a continuous form. Section 2.1, 2.2 and 2.3 have been revised for the detailed description.

**line 158**

--------

*c was defined as a vector, not as a continuous function.*

Thanks for the correction. $I(f)$ has been used to represent the seminorm defined on a continuous function $f$. We have revised section 2.3 for the revised definition.

**line 159**
* * *
*What items in which summation? There is an integral in (7).*

The discrete and continuous formulated have been distinguished. The discrete total squared curvature has been added. Eq. (11) and (12) in section 2.3 have been revised for the definitions.

**line 160:**
* * *
*"This is how the LTD algorithm does to add additional(sic!) equations?" What does this mean?*

Two additional linear equations are introduced at each pixel defined through the third-derivate prior information. There will
be $2N_c$ linear equations appended to the original linear equations defined by the PIC observations, resulting in a new system of linear equations which has $(2N_c + N_b)$ equations and $N_c$ unknowns, where $N_c$ is the pixel number, $N_b$ is the beam number. Assuming the new augmented kernel matrix is $A$, observation vector is $p$, then the new system of linear equations is $Ac=p$. We have revised the description in the section 2.3.

**line 161:**
* * *
*Is such a complicated equation really more efficient computationally than two much simpler ones? What are the involved algorithmic complexities?*

The LTD and MC algorithms use the same approach (the NNLS optimization algorithm) to solve the generated system of
linear equations. The main different is the number of equations. The MC algorithm improves the computational efficiency by reducing the equation number by half comparing to the LTD algorithm.

**line 165:**
* * *
*It is unclear why this biharmonic equation is necessary. You are already minimizing a cost function in (6). Equation (7) gives you an immediate way to calculate |c| required by (6). Computing discretized ddc/ddx+ddc/ddy and computing the Euclidian norm should give you (6) without the need for higher*

*derivatives in the definition of the problem. To efficiently solve (6) one might need higher derivatives depending on the chosen algorithm (e.g. Gauss-Newton), but your paper does not detail this part very well.*

*Please describe in detail by which algorithm (6) is solved and how (9) plays a role in that.*

The reviewer is correct. Solving the biharmonic equation gives the same solution to the minimization problem. But we did not find the solution in this way. Instead, we used the equations derived from the minimum curvature principle. To minimize the seminorm, the partial differential of the total squared curvature with respect to the pixel concentration needs to zero for each pixel, resulting a difference equation. A new system of linear equations is set up based on these difference equations.
The inverse problem was solved by the NNLS optimization algorithm. We have removed the definition of the biharmonic equation. The detailed description of the derivation of the equations can be found in section 2.3.

**line 169:**
* * *
*Again, the regularization weight typically depends on diffusion coefficients and measurement errors and is often a tuning parameter. The grid size should directly be implemented in the finite difference equations.*

The grid size has been implemented in the finite difference equations. A weight needs to be assigned to each equation depending on the uncertainty of the observation. Assuming the analyzers have the same performance, the uncertainty is mainly related to the path length. Therefore, equations are assigned weights inversely proportional to path length to make
sure different paths have equal influences. In this study, the lengths of the laser paths are approximately equal to each other. So their weights were all set to be the same value which is scaled to be 1. The weights for the third-derivative prior equations were assigned to be the same value of w because they were all based on the same grid lengths. *w* was determined in the same way of determining the regularization parameter μ in the Tikhonov regularization

The regularization parameter determines the balance between data fidelity and regularization terms. Determination of
optimum regularization parameter is an important step of the regularization method. But the regularization parameter is problem and data dependent. There is no general-purpose parameter-choice algorithm which will always produce a good parameter. For simplicity, we use the method based on discrepancy principle. The regularization parameter is chosen from a finite section of a monotonic sequence. For each value of μ, an optimal solution is derived by solving the inverse problem. Then the discrepancy can be calculated. The regularization parameter is determined to be the highest value that makes the
discrepancy equal to $N_b\sigma^2$, where $N_b$ is the number of beams, $\sigma$ is the standard deviation of the noise. In this study, the reconstructions varied only slowly with the regularization parameters. Therefore, precise selection of the parameter was not necessary. The regularization parameter was selected from four widely varying values. The one produced the smallest discrepancy was used.

We have added a separate paragraph at the end of section 2.3 to discuss how the weights and regularization parameter are
determined.

**line 180:**
* * *
*What interpolation applied after the reconstruction process? The pixel-based algorithm assumes constant values over* 630 *constant pixels. There is no smoothing interpolation, which would not deteriorate the fit to the measurements, i.e. deteriorate the results.*

What we want to explain here is that the variational approach is another way to find the interpolating splines. Therefore, by solving the minimization problem with the smoothness seminorm penalty, we will get a smooth solution similar to the result of applying spline interpolation in the view of the forward model. The final solution is smooth and also has characteristics 635 related to the spline interpolation. But the "interpolation" is achieved during the inversion process instead of after it, which is a key to improve the reconstruction accuracy. This is important because an interpolation after the reconstruction cannot correct the error resulting from the reconstruction based on coarse spatial resolution. We have rewritten section 2.3.

**line 186**

--------

*Here c is defined, for the first time, as a continuous function! Please properly distinguish the "different" c's.*

Thanks for the correction. $g(x, y)$ has been used to represent the continuous concentration distribution.

**line 189**

--------

*What source number? (10) defines only a single source. If you use multiple sources, please accommodate this in (10).*

The source number varies from 1 to 5. For multiple sources, the resulting concentration distribution is the superposition due to each source. We have revised the description in section 2.4 to clarify this.

**line 189**
* * *
*You state that the peak width was set randomly. Was it chosen randomly from the listed peak width of line 187f?*

Yes. The values were randomly chosen from the predefined set. We have updated the description to clarify this.

**line 195**
* * *
*You were using c_i,j above for the 2-D fields, why now only c_i?*

The pixel concentration can be arranged both in 1-D and 2-D forms. We now distinguish these two arrangements by using $c_j$ and $C_{k,l}$, where $j$ is the index of a 1-D vector $c$, $k$, $l$ is the row, column index of the 2-D matrix $C$ respectively. The 1-D form is used for the linear equations as unknown vector. The 2-D form is used for the finite difference operations. The revised definitions are in section 2.2.

**line 201**
* * *
*How was the location of the highest peak located?*

The peak is located by searching the largest concentration value on the map. When there are multiple locations having the same values, the centroid of these locations is used. We have revised the manuscript to describe this.

**line 210**
* * *
*Why did you not apply the other algorithm on your fields for better comparability?*

This study mainly focuses on the comparison between the new MC algorithm and the LTD algorithm because they have similar formulas. While MG-GT algorithm uses a different approach. The detailed comparison with the MG-GT algorithm may be conducted in another study with the purpose to investigate their performance in different applications.

**line 215**
* * *
*While it seems to work, the pixel based algorithm derives pixels, not a continuous field. It is straightforward to derive a spline interpolated field directly, if desired for the higher accuracy. One simply needs to compute the integrals over the spline interpolated field for the coefficients when computing the error to the measurements. This can be accomplished by a linear matrix multiplication. I expect this to deliver similar results as the other methods at maybe even faster speed due to the smaller number of involved equations. Please discuss the choice of your simpler forward model.*

With the purpose of achieving a smooth reconstruction, there is an important difference between applying interpolation after the reconstruction and applying smooth regularization during the reconstruction process. In the former situation, high-resolution grids cannot be used in order to make the inverse problem well-posed. The resulting solutions have coarse spatial resolution. Large error may occur and cannot be improved by the interpolation after the reconstruction has been done. In the latter case, high-resolution grids can be used during the reconstruction process. The MC approach evaluates the discrepancy based on the high-resolution values that are the same as the reconstruction outcomes. Errors due to coarse spatial resolution are corrected during the process. We have revised the explanation at the end of section 2.3.

**line 243**
* * *
*Why does the necessary computation time scale with the number of sources? Shouldn't it be proportional to the problem size?*

It implies that the underlying distribution also affects the computation. Because when the source number is small, most of the area has almost zero values, which saves time to find an optimal solution. As the source number increases, the underlying concentration becomes complicated. It will need more iterations to approach an optimal solution. We have added a new section of 3.6 investigating the influence of the grid size. The result shows that the computation times for the LTD and MC algorithms exhibit approximately exponential growth trends with the increasing of the pixel number. While the LTD algorithm has a faster increasing rate than the MC algorithm. We have revised the discussion in section 3.6.

**line 283**
* * *
*"oversmooth issue" - necessarily, the amount of information cannot increase between the measurements and the solution. Due to the chosen regularization, the result will be necessarily smooth. If it is "oversmooth" depends on whether the a priori assumption of smooth fields is correct or not.*

It is true that this issue depends on the actual underlying distribution. Actually, the underlying distribution has considerable influence on the performance of a reconstruction algorithm. In this study, we have adopted the variational interpolation approach to the tomographic reconstruction. As a result, now we can better understand the physical interpretation and the characteristics of applying the smooth regularization to the inversion according to the close connection with the spline functions. As a consequence, the issues related to the spline functions may also happen in the tomographic reconstructions and need to be further investigated.

*In case that this assumption does not hold, "better" (less smooth) results can be achieved by Total-variation minimization (isotropic or anisotropic) and primal dual methods, e.g. Split-Bregman. I doublt this would fit better to your problem, though.*

Actually, we have tried the TV minimization. But it did not give the smoothness effect we expected. We think it is because the observations (line integrals) are too less to sufficiently distinguish the concentration gradient. As a result, an approach providing sharp gradient may give physically unrealistic solutions. However, this result is based on the specific synthetic problem in this manuscript. If different beam configurations (more beam number with optimized geometry) were used, the TV approach might be a good try.

**MINOR REMARKS**

=====

**line 55**
* * *
*"question". This is called an "inverse problem".*

corrected.

**line 74**
* * *
*grids -> grid points?*

The "pixel number" is used.

**line 85**
* * *
*necessary -> necessity*

"need" is used.

**line 144**
* * *
*-> "third-order forward difference operator"*

corrected.

**line 160**
* * *
*Which "multiple items"?*

This section has been rewritten and this sentence was removed.

**line 174:**
* * *
*"For pixel-based"->"For conventional pixel-based", posted->posed*

corrected.

**Response to comments of reviewer 3**

We thank the reviewer for the helpful comments and suggestions, and for careful reading of the manuscript. Listed below are our responses, with the original comment/question displayed in italics.

**Anonymous referee report**

*The paper from Sheng Li and Ke Du proposes a new minimum curvature (MC) algorithm to apply smoothness constraints in the tomographic inversion of optical remote sensing measurements, to reconstruct the spatial distribution of atmospheric chemicals in a given domain (a 40 x 40 m square area in the example given). The authors compare the performance of their new proposed method to that of other existing methods, such as the non-negative least squares (NNLS) and the low third*
*derivative (LTD). The performance is assessed on the basis of a few test maps containing one or several (up to five) bi-variate Gaussian sources. Apparently, the MC algorithm performs significantly better than the NNLS method, and shows almost the same performance of the LTD algorithm in terms of reconstruction accuracy. Compared to this latter, however, the MC algorithm allows to save from 27 to 35% computation time, depending on the number of sources in the domain.*

*The subject of the paper is interesting, comprehensively presented in the introduction and put in the context of the existing*
*literature on the topic. The method used for the assessment, however, is not sufficiently general and could be improved. The presentation of the algorithms assessed is not sufficient, the actual equations used should be included in the paper. Regarding the language of the text, I am not native English speaker, thus I cannot provide a reliable feedback. However, the language sounds a bit "strange" to me at several instances. Therefore I recommend a review by a language Editor. Also, I would suggest to avoid flooding the text with acronyms. Several of them are not really necessary and make reading the*
*paper uncomfortable.*

*In conclusion, I am very sorry but I can recommend this paper for publication in AMT only after major improvements, as outlined in the comments below.*

Thank you for the comments and suggestions. We have done a major revision according to your suggestions. Please see the responses below for specific changes we have made to the manuscript. The language has been edited by a highly experienced
and native English-speaking editor.

**General comments**

*My main concern is that the authors have compared the field reconstruction errors of the NNLS, LTD, MC (and GT-MG) algorithms on the basis of a set of only five test distribution maps. I would say that they have verified some necessary*
*conditions which, however, are not sufficient to assess the relative efficiency of the considered methods. Each solution depends both on the measurements and on the constraints applied. Here it is not clear whether the LTD and MC solutions perform better than the NNLS because of the smarter applied constraints or because of the specific experimental distributions (bi-variate Gaussians) considered in the examples given.*

There are several things which need to be mentioned about this comparison.

(1) The performance of the algorithms is problem dependent. The beam configuration and underlying concentration distribution also affect the reconstruction quality. There is no one algorithm that performs absolutely better than others all the time. Even the NNLS algorithm could give the best result under some specific conditions (Wu and Chang, 2011). This is the common situation for an optimization algorithm.

(2) The bi-variate Gaussians are good choice to simulate the diffusion process. It has been commonly used in the
comparisons of the algorithms in the ORS-CT problems. The generated maps better simulate distributions which have several hotspots. It is true that the real distribution map may be complicated depending on the specific dispersion conditions. Other methods to generate the test maps can also be used. For example, the bivariate lognormal distribution was used in Wu and Chang (2011) to simulate a non-symmetric distribution. But that only represents one situation, and the distribution function may not be better than others in other cases.

(3) Another reason that the test maps were generated in this way is for the purpose to compare with the results from the GT-MG algorithm, which used the bi-variate Gaussians. And the beam configuration was also set to be similar to that in the GT-MG approach. This is a common set for comparisons of tomographic reconstructions and was used in practical applications (Dobler et al., 2017).

*Error covariance matrices and averaging kernels (see e.g. Rodgers, 2000) are broadly used tools in the atmospheric remote sensing community to characterize the recovered spatial distribution (yes, also 2D distributions ...) from the point of view of the retrieval error, and of the spatial resolution (width of the Point Spread Function) of the measurement chain (measuring plus inversion systems). Applying these tools to the inversion methods considered in the paper is possible, thus the authors should use them. For example, from the analysis reported in the paper, it is not self evident that the spatial resolution of*
*their measuring system changes strongly depending on the x,y position within the squared field considered: there are grid-elements which are crossed by 2 or 3 beams, and others (near the sides of the field) which are not sounded at all. Thus, the spatial resolution must be very poor near the sides of the squared field and much better near the center. I believe this feature would be self-evident from maps of the diagonal elements of the 2-dimensional averaging kernels of the different solutions considered (see e.g. von Clarmann et al, 2009).*

Thanks for the good suggestions. We have added the averaging kernel matrix into the manuscript to determine whether the concentration can be independently predicted and how regularization limits reconstruction accuracy. In the 2-D tomographic reconstruction, the averaging kernel is considerably affected by the beam geometry and is better to be used as a measure to evaluate the beam configuration. But it also reflects the regularization error given the same beam geometry in this study. We used a measure called fitness which was defined as the average Frobenius distance between the resolution matrix and the
identity matrix to predict the reconstruction error. The MC algorithm shows slightly better performance than the LTD algorithm with a fitness value of 1.3878 comparing to 1.4411.

The off-diagonal elements are not zeros. The reconstructed concentration at each pixel is a weighted average of the concentration of the surrounding pixels because of the smoothness regularization. Each row of the resolution matrix can be regarded as smoothing weights. The 2-D display of the row of the averaging matrix gives a clear dependence of the beam
configuration, while the diagonal elements may not provide much information in this case.

Section 2.5 and 3.5 have been revised for the detailed description.

**Specific comments**

*Lines 42-44: please include references for the mentioned techniques. They are not standard for the whole atmospheric*
*remote sensing community.*

The reference has been added (Radon, 1986; Herman, 2009; Censor, 1983).

*Lines 73-74: not only, I guess. The chosen pixels should be crossed at least by one beam, otherwise the NNLSF is ill-posed.*

That is correct. For the traditional pixel-based methods, the pixel is required to be passed by at least one beam, otherwise the problem will be ill-posed. The regularization technique is the method to solve the ill-posed problems by adding *a priori* information into the original problem. This is shown in this study by using high-resolution grid division, in which case many grids are not passed by the laser beams. But we still got a good reconstruction through smoothness regularization.

*Line 122: I would name the small squares as "pixels" instead of "grids".*

"pixels" has been used.

*Line 127: if the PIC is measured at the retro-reflectors, then you have only 16 measurements, or 20 measurements if retro-reflectors are installed also at the corners of the square. Instead, I guess that for the NNLSF you need at least 36 measurements. Please make an effort to describe more thoroughly the experimental setup.*

It is correct that there were 20 retroreflectors installed (including the ones at the corners). We can see from bean configuration (Fig. 1) that each retroreflector reflects two beams coming from different directions. Therefore, the total beam number will be 20×2=40. After removing the overlapped beams along the diagonals, the total beam number is 38. We have revised section 2.1 to give a more detailed description of the setup.

*Section 2.1: it would be interesting if the authors could explain the details of the experimental setup, I could not understand which is exactly the measured quantity. This would be useful also to understand to which degree the linear formulas (1) and (2) are accurate.*

For each laser beam, the path-integrated concentration (PIC) is measured by the analyser. The predicted PIC for one beam equals to the summation of the multiplication of the pixel concentration and the beam length passing it. In general, let us
assume that the site is divided into $N_c=m\times n$ pixels, which are arranged as a vector according to the left-to-right and top-to-bottom sequence and indexed by $j$. The average concentration for $j$-th pixel is $c_j$. The total number of laser beams is $N_b$ which are indexed by $i$. The length of the laser beam passing the $j$-th pixel is $L_{ij}$. Then, for the $i$-th beam, the measured PIC $b_i$ is contributed by all the pixels that it passes. A system of linear equations can be set up for all the beams: $b=Lc$, where $L$ is the kernel matrix that incorporates the specific beam geometry with the pixel dimensions, $c$ is the unknown concentration vector
of the pixels, $b$ is a vector of the measured PIC data.  This section has been rewritten to give a detailed explanation of the experimental setup.

*Equation (3) assumes that all the measurements have the same precision, which may not be the case if the signals observed are very different in intensity (e.g. because of the different absorption paths). Could-you please add a comment?*

Yes. A weight needs to be assigned to each equation depending on the uncertainty of the observation. Assuming the analyzers have the same performance, the uncertainty is mainly related to the path length. Therefore, equations are assigned weights inversely proportional to path length to make sure different paths have equal influences. In this study, the lengths of the laser paths are approximately equal to each other. Thus, their weights were all set to be the same value which is scaled to be 1. The weights for the third-derivative prior equations were assigned to be the same value of $w$ because they were all
based on the same grid length. Section 2.2 has been revised for the revised description.

*Line 139: With equation (3) you require a solution with "small" $Li\ \mathbf{c}$ norm, whereas, usually one requires a small $Li\ (\mathbf{c} - \mathbf{ca})$ norm, where $\mathbf{ca}$ is some prior estimate of $\mathbf{c}$. Please explain the rationale behind your choice of $\mathbf{ca} = 0$.*

Using $(\mathbf{c}-\mathbf{ca})$ means that we have some predefined values $\mathbf{ca}$ which we use to regulate the solutions to make it close to the
prior estimations. This is the case in the atmospheric sounding problem, where some modeled values are used as the prior estimations. But for the reconstruction of the 2-D concentration distribution of arbitrary area site. There is no such prior information about the underlying concentrations. The only information we know is that the values are non-negative.

*Line 148: If the regularization parameter $\mu$ of eq. (3) is grid-dependent, then I would expect it to appear in some vector form*
*in equation (3) rather than as a scalar. How do you establish the actual value of $\mu$? Which is the solution of the LTD algorithm? Please specify the equation.*

We have revised the equation. Now the grid length is part of the third-derivative prior equation. The regularization parameter is independent of the grid size and determines the balance between data fidelity and regularization terms. Determination of optimum regularization parameter is an important step of the regularization method. However, the regularization parameter
is problem and data dependent. There is no general-purpose parameter-choice algorithm which will always produce a good parameter. For simplicity, we use the method based on discrepancy principle. The regularization parameter is chosen from a finite section of a monotonic sequence. For each value of $\mu$, an optimal solution is derived by solving the inverse problem. Then the discrepancy can be calculated. The regularization parameter is determined to be the highest value that makes the discrepancy equal to $N_b\sigma^2$, where $\sigma$ is the standard deviation of the noise. In this study, the reconstructions varied only slowly
with the regularization parameters. Therefore, precise selection of the parameter was not necessary. The regularization parameter was selected from four widely varying values. The one produced the smallest discrepancy was used. Section 2.2 has been revised for equations and description.

*Line 149: In principle, the NNLS algorithm does not use constraints, correct? Here you are explaining the LTD algorithm,*
*so it cannot be solved with the NNLS approach. Maybe you refer to the Newton method? Please explain.*

The non-negative least square (NNLS) optimization algorithm is the first widely used optimization method proposed by Lawson and Janson (1995) in their textbook to solve the non-negative constrained least square problem. In this paper, the "NNLS algorithm" to the tomographic reconstruction refers to solve the original system of linear equations by using the NNLS optimization algorithm without adding other *a priori* information. The LTD approach generates a new system of
linear equations with non-negative constrains, which can also be solve by the NNLS optimization algorithm. We have revised section 2.1 for the description.

*Lines 158-165: Up to eq.(6) and later also in eq. (9), $\mathbf{c}$ is a vector. In eq.s (7) and (8) "c" seems a function. Please improve the notation, it would be difficult to implement your MC algorithm based on your description.*

We have revised the formulates in the manuscript to give a more accurate definitions of the quantities and distinguish the continuous and discrete functions. The seminorm is defined as an integral of a continuous function, which is the total squared curvature in the MC algorithm. Then, it is discretized to be the sum of the squared curvature at each pixel. Also, *c* is defined as a 1-D vector of the unknown pixel concentrations, while *C* is defined as a 2-D matrix of the pixel concentrations when they are arranged in the 2-D form. *f(x,y)* is used to describe the underlying concentration distribution. Section 2.1, 2.2, and 2.3 have been revised for the detailed definitions.

*Line 167: I have understood that you are finally using eq.(9), that is the discretized form of eq.(8). Is eq.(9) more or less equivalent to eq.(7)? This description is very confusing.*

The seminorm is defined as the total squared curvature in the MC algorithm. It is first discretized to be the sum of the squared curvature at each pixel. Then, in order to minimize the seminorm, its partial derivative with respect to the concentration at each pixel needs to zero, resulting in a difference equation. This equation is appended at each pixel as a smoothness prior equation and a new combined system of linear equation is set up. We have revised section 2.3 for the equations and the detailed description.

*Line 169: the same comment I made for $\mu$ (line 148) here applies to $\omega$. Which constant for the inverse proportionality did you use?*

The weight *w* is determined in the same way as determining the regularization parameter of $\mu$, which was described in above answer for question of line 148. We also revised section 2.2 for the detailed description.

*Line 187: If c(x,y) is a concentration, then Q cannot be measured in ppm (that is a mixing ratio). Eq.(10) does not contain $\sigma$, it contains $\sigma x$ and $\sigma y$…*

The ppm unit has been changed to mg/m3. $\sigma x$ and $\sigma y$ have been defined.

*Lines 212-213: this sentence is not clear to me.*

This sentence has been revised as "They did not measure the peak location error and used a different way to calculate the exposure error by limiting the calculation domain in a small area near the peak instead of the whole map. Therefore, the results of the GT-MG algorithm are listed as a reference and only the measure of nearness is compared. "

*Line 220: "The smaller the number of sources, the better the reconstruction quality". I think this is intrinsic to the definition*
*of the nearness quantifier. Please comment.*

There is a mistake in the sentence. It has been revised as "The smaller the nearness value, the better the reconstruction quality". The trend of nearness with the increase of the source number is discussed in that paragraph.

*Lines 231-233: do you mean that the "peak-location" quantifier could mistake peaks with similar amplitudes? In this case it*
*would be advisable to refine eq.(12) or to apply it with some caveats.*

When there are two or more peaks on the map whose peak magnitudes are comparable to each other, the reconstruction algorithm may not identify the correct location of the highest peak. Therefore, a large error may happen when the highest value on the reconstructed map is located on the wrong peak.

*Line 233: I think that, as it is, eq.(12) is reliable only if the peaks to be reconstructed have amplitudes that differ from each other by much more that the error with which they are retrieved. Why the "source number" counts so much?*

The reason is given in the previous answer. The large error is mainly because the reconstructed highest peak is the wrong one. When there are more peaks, this kind of error is more likely to happen.

*Line 239-240: I am skeptical about your general statement regarding the NNLS performance. I would suggest adding some details regarding how you actually computed the NNLS solution.*

Actually, the NNLS algorithm is used as one representing method to solve the unregularized inverse problem. The main performance difference is due to the unregularized and regularized reconstructions. The unregulated inversion uses coarse grid division and produces fewer pixel concentrations representing constant values in the grids. Therefore, this method has
difficulty dealing with the large variation in a scale smaller than the grid size. It fits the true distribution better when the distribution becomes more uniform. We have updated the manuscript to include this explanation.

**Technical corrections**

*Line 50: summarising ??*

This sentence has been revised as "The path integral is approximated by the summation of the product of the pixel value and the length of the path in that pixel."

*Line 85: necessary ?*

The word has been replaced by "need".

*Line 136: what is the superscript "21" ?*

The number has been replaced with a reference.

*Line 174: "well-posed"*

Corrected.

*Line 212: it (?)*

Corrected.

*Line 224: maybe "complexity" ?*

Corrected.

*Line 228: are slightly better ...*

Corrected.

*Line 241: derivation ??*

It is the standard deviation of the results from 100 tests.

**References**

*Rodgers, C. D.: Inverse Methods for Atmospheric Sounding: Theory and Practice, in: Series on Atmospheric, Oceanic and Planetary Physics, Vol. 2, edited by: Taylor, F. W., World Scientific, 2000.*

*von Clarmann, T., De Clercq, C., Ridolfi, M., Hoepfner, M., and Lambert, J.-C.: The horizontal resolution of MIPAS, Atmos. Meas. Tech., 2, 47–54, https://doi.org/10.5194/amt-2-47-2009, 2009.*

Dobler, J. T., Zaccheo, T. S., Pernini, T. G., Blume, N., Broquet, G., Vogel, F., Ramonet, M., Braun, M., Staufer, J., Ciais, P.: Demonstration of spatial greenhouse gas mapping using laser absorption spectrometers on local scales. Journal of applied remote sensing. 11(1), 014002, doi: 10.1117/1.JRS.11.014002, 2017.

Wu, C.F., Chang, S.Y.: Comparisons of radial plume mapping algorithms for locating gaseous emission sources. Atmospheric Environment, 45, 1476-1482. DOI: 10.1016/j.atmosenv.2010.12.016, 2011.

---

## Referee Report (RR1)

**General**

As compared to the first version, the paper by Sheng Li, Ke Du has been significantly changed. Many of the concerns I pointed out in the first review have been addressed. I apologize that while reading the new version of the paper, unfortunately, I found some new relevant issues. Some of them are crucial and, to my opinion, should be carefully considered before publication.

In general the language has been greatly improved, however, I would recommend the authors to double check if the editing process has *always* conserved the original meaning of what they wanted to say.

**Specific comments**

Sect. 2.1: the section has been significantly improved. I see only two remaining issues:

a) Linearity of equation (2). Your statement at line 848 of your replies *"The predicted PIC for one beam equals to the summation of the multiplication of the pixel concentration and the beam length passing it"* holds only in the "optically thin" regime, i.e. only when a small fraction of the laser signal is absorbed by the pollutant to be measured. If the pollutant concentration is large and / or the medium is not very transparent at the laser wavelength, then for sure linearity does not hold. Since the laser wavelength can be tuned, usually it is possible to find a suitable wavelength at which, for average or expected pollutant concentrations, the medium is sufficiently transparent to fall in the linear regime. To my opinion it would be convenient to state explicitly that you are working in this hypothesis.

b) Equation (3): is it really needed to require $c \geq 0$ ? Assume that the real concentration of the pollutant to be measured is zero. If you obtain the concentration as the average of several measurements, then, because of the analyzer's noise, these measurements should be evenly distributed around zero to give a zero average value. If you constrain the solution $c$ to be greater or equal to zero, then you remove the measurements that otherwise would be negative because of the measurement noise. Thus you introduce a bias (or exposure error) in the average. Of course you don't see this type of error in the tests presented later in the paper, because the test 2D distributions considered do not include a "zero concentration" case, and because all your formulas fully ignore the measurement noise error (that is the noise error on your $b$ vector).

Sect. 2.2: If I understand properly, in practice what you call LTD method is the Tikhonov regularization using the discrete 3rd derivative operator $D_3$. You tune the regularization strength $w$ using the discrepancy principle. Then I have a question: you can not introduce the two equations (4) at every pixel of your domain, as you would have problems at the edges of the domain (see index values k+2, k+1 and k-1 in eq. 4). Thus, you can actually introduce only $2N_c - 6$ additional equations (not $2N_c$ as you state at line 160). If you do so I think you get in trouble if one or more pixels of your domain are not crossed by a ray path. Actually, in this case $L^t L$ would be singular and, if also $T^t T$ is singular, then you cannot invert the matrix $A^t WA$ mentioned at line 172. In conclusion, I think you are using some extra constraints (also involving 0-th order derivatives) at the edges of your domain (as you seem to suggest in the text after eq. 7). Please clarify. Also, what are the symbols $m$ and $n$ in eq. 7?

Line 243: please explain clearly the equations you use at the edges of the domain, see also the analogous comment above.

Line 245: requiring $c \geq 0$ introduces a bias (i.e. a systematic exposure error) for very small values of the real concentration distributions $c$, see also the analogous comment above.

Sect. 2.4: Here you define the "true" concentration distributions that will be considered later to assess the various reconstruction methods. You should say also something regarding the synthetic observations $\mathbf{b}$ that you build on the basis of g(x,y). Which is the spatial mesh (or grid) that you use for the calculation of $\mathbf{b}$, starting from g(x,y)? Do you add pseudo-random measurement noise to $\mathbf{b}$, to emulate the real world situation in which the instrument is subject to measurement noise? Your equations discard the measurement error. Mathematically this is equivalent to assume uncorrelated noise equal to 1, thus you should add to each element of $\mathbf{b}$ a pseudo random noise with zero average and standard deviation equal to 1.

Lines 293 and ff: the unusual naming conventions used in this part make it very difficult to understand for me and, I guess, also for the whole atmospheric community. $\mathbf{G}$ matrices defined at lines 298-300 are usually called "gain matrices". It is true that in this case they also represent the "generalized inverse" of $\mathbf{L}$ because the forward model $(\mathbf{Lc})$ is linear. Despite of that, I would still call $\mathbf{G}$ the "gain matrix", just for uniformity with Rodgers 2000. Equations at lines 299 and 300 should contain a "+" sign within the parenthesis. Matrix $\mathbf{R}$ is usually called "averaging kernel", not "resolution matrix", just because it shows how a change in the real concentration field maps onto the the the concentration estimated by the inversion system. The "resolution" is usually a scalar or vector quantifier computed on the basis of $\mathbf{R}$. Line 304: what you call "regularization error" is usually called "smoothing error". Line 305: what is the perturbation error $\mathbf{G}$ delta_b ? Why should the observed quantities be perturbed? Maybe delta_b is the measurement error (noise + calibration, etc.)? Please explain. Line 307: note that the non - sensitivity to the "perturbation error" is not always an appreciated property for the solution. In fact this means that the solution is mostly independent from the observed quantity, therefore it mostly depends on the applied constraints, that is exactly what one usually wants to avoid. Eq. 23: why don't you write explicitly the expression for the Frobenius norm? Which is the advantage of using the Frobenius norm over the more usual trace[$\mathbf{R}$] that, according to Rodgers, 2000 would represent the number of degrees of freedom for the signal? If $\mathbf{R}$ is not a orthogonal matrix (like in this case, I guess) then I am not able to find an easy interpretation for your quantifier of eq.(23).

Line 365 and ff: how do you interpret the systematically slightly worse performance of MC as compared to LTD in terms of exposure error? I am also surprised to see such a large exposure error in the NNLS method as compared to the other two methods. Maybe this due to the larger size of the pixels used with NNLS? In this case it would be better to change eq.(20) by multiplying each concentration by the area of the pixel to which it refers (actually eq.(20) should compare the integrals of the concentration in the considered domain).

Sect. 3.4: at line 144 of the revised paper finally you reveal that you are using the "lsqnonneg" MATLAB routine to minimize the cost functions relating to the various methods considered. I was not able to establish what is exactly the minimization method used by this MATLAB function, however, for sure it does not find the solution mentioned at line 172 of your revised manuscript, because this latter is not bound to be positive. As explained later, I suspect that lsqnonneg is using iterations, which is not optimal at all for the linear case (b = Lc) you are dealing with. This implies that the run-time analysis presented in sect. 3.4 is applicable only to the unlucky case in which one uses lsqnonneg. The optimal solution of your inversion would be to implement, directly, in a computer program the matrix operations at line 172. This would make the computation time dependent only on Nb (number of measured PICs) and on Nc (number of unknowns). Both Nb and Nc do not depend on the number of sources that you put in your domain, thus the computation time would not depend neither on the number of sources considered nor on their amplitude and location in the domain. You also claim that the MC method is much faster with respect to the LTD because of the fewer constraint equations. To my experience, if the solution at line 172 would be implemented, most of the computation time would be spent in the inversion of matrix $\mathbf{A^t WA}$, of

dimension Nc. The number of constraint equations used would impact only the calculation of the product $T^t T$ (that is needed to compute $A^t WA$), thus the total computation time should depend only very marginally on the number of constraints used.

Another discussion point are the absolute values of the computing times shown in Table 4 and their standard deviations: why the computation time changes and thus shows a standard deviation also when the method used and the number of sources do not change? (I suspect this is because lsqnonneg uses iterations whose number changes from test to test). Why is computation time so long? Are you using a very old CPU? Please specify. Assuming your inversion problem dimensions, finding the solution given at line 172 would require less than 1 second with a standard CPU.

In conclusion, finding the solution as $c = (A^T W A)^{-1} A^T W p$ (as specified at line 172) would be much faster and the savings in computing time achieved with the MC method over the LTD would be much less important.

Line 378: what do you infer from fig's 4a and 4b ? Are they useful? I was not able to extract information from them. In my previous review I was suggesting to include only Fig.s 4e and 4f. Of course additional figures are welcome, however they should convey useful information that should also be discussed in the text.

Line 388: figures 4e and 4f show diagonal elements significantly smaller than 1, this means that your regularization is actually very strong ($w$ is very large). What do you get for trace[$R$] / Nc ? I guess it is << 1. In practice I feel that with a much softer regularization you would get solutions with better accuracy and similar smoothness.

Figure 4: at least, please use the same color scale in the left and right maps, otherwise the comparison of the two methods is very hard ….

Section 3.6: the results reported in this section are not useful as they are not general for the experimental setup considered, they depend on the dimension of the assumed sources. From what stated at line 399, I understand that these sensitivity tests are performed assuming 5 Gaussian source functions, whose minimal Full Width at Half Maximum (FWHM) is 2.8 m * 2.355 = 6.59 m (see line 272 and the properties of the Gaussian function). When the dimension of the smallest source function is about 6.6 m, clearly there is no much advantage to go from a 18x18 grid (pixel size = 40m / 18 = 2.22 m) to a 40 x 40 grid (pixel size = 40m / 30 = 1.33 m) as in both cases the pixel size is significantly smaller than the source size. Usually, in real situations, the size of the source is unknown and what is useful to know is the spatial resolution of the measuring plus inversion system. The averaging kernels are usually an adequate tool to answer this question: the dimension of the main yellow spots of maps in fig. 4c and 4d quantify the spatial resolution of your measuring system at the location of indexes 16 and 4.

Lines 410 – 413: from the theoretical point of view, the growth of computation time for this inverse problem should roughly follow a polynomial law as a function of the number of pixels. Did you really prove that the behavior you find is exponential? If the behavior is really exponential then you should give a justification so as why the computation time increases so rapidly in your case.

Conclusion section: as outlined above, in my view the time computing savings of the MC method, if any, should be assessed using an optimal solving algorithm for your inversion problem, not the MATLAB lsqnonneg routine. In conclusion, either you renounce claiming the time savings or you implement an optimized solver (like eq. at line 172 implemented in FORTRAN or in C).

**Minor corrections**

Line 14: I would say "regularization term" (regularization is not a factor in the inversion formula)
Line 22 (and line 113): It is also simpler to perform... maybe you meant "implement" instead of "perform" ?
Line 204: bending ?
Line 272: units of sigma_x and sigma_y should be "m" (meters).
Line 416: seminors ?

---

## Author Response (AR2)

**Response to comments of reviewer 2**

We thank the reviewer for the helpful comments and suggestions, and for careful reading of the manuscript. Listed below are our itemized responses, with the original comment/question displayed in italics.

**GENERAL COMMENTS**

====

*The paper describes a minimum curvature based regularization schemed for deriving 2-D trace gas concentrations from optical remote sensing and tomography. The chosen regularization scheme is sensible and the method seems to be an improvement over the state of the art in the field.*

*The topic fits the journal.*

*The textual description is good and the different methods are well introduced.*

*The paper may be published after addressing the following minor issues.*

Thanks for the comments. We have revised the manuscript according to your comments. Please see the responses below for specific changes we have made to the manuscript.

**SPECIFIC COMMENTS**

=====

**Eq. 4**
* * *
*The finite difference operator requires the division by the third power of the grid distance. The results are likely correct as one needs to multiply with \delta x \delta y in Eq. (5) to properly approximate the 2-D integral over the third order derivative. However, in case \delta x unequal to \delta y, an error would be made.*

Thanks for the correction. We have changed the division to the third power of the grid distance in Eq. 4 to make it more accurate.

**Eq. 7**
* * *
*In what way are the matrices T and D_3 similar/different?*

Matrix $T$ contains the third derivatives of all the pixels in both $x$ (row) and $y$ (column) direction. For the derivatives in $x$ direction, the matrix is defined by $D_3$. For those in $y$ direction, each row of the matrix still contains the coefficients [-1 3 -3 1], but they are separated by $n$ (the number of pixels in $x$ direction) "zero" elements to correctly define the difference in the y direction, because in $y$ direction, the difference of index numbers of two neighbor pixels is $n$.

**Eq. 8**
* * *
*Here the fourth symbol is introduced for a weight factor after w, W, and μ. Is this necessary, or can one simplify the employed notation?*

40  They all can be viewed as weight factor in this study. But they may have different interpretations in different fields. For example, the $w$ in the LTD approach represents the weight for a linear equation and was not described as regularization parameter. $W$ is a matrix containing the weights for all linear equations. The weights for different equations can also be different. $\mu$ is a single value to determine the weight of the regularization term in the regularized inversion problem. In this view, $\omega$ in Eq. 8 and Eq. 10 and $\lambda$ in Eq. 16 are also regularization parameter and have been replaced by $\mu$ according to the
45  reviewer's suggestion,

**Eq. 10**
* * *
*Here, a needless approximation error is introduced via the c_i. You derive, effectively, a continuous representation f of the*
50  *2-D distribution, which could be used for a better "forward model" F mapping the continuous distribution f onto measured values b. You might think about being more general here before justifying your choice of forward model based on the pixel-based approach.*

Thanks for the suggestion. In a general case, the *PIC* of a path is a line integral over concentration field. We have added this integral formula to the manuscript.

55

**Eq. 12**
* * *
*The grid distance is missing in the discretized integral. Please differentiate. Also, I is now both a function of f and a sum. As the "new" I is differentiated later, it would be good to introduce it as a function here as well.*

60  Thanks for the correction. The grid distance has been added to the equation. To distinguish continuous and discrete form, we have changed the discrete total squares curvature $I$ to $S(C)$, which is a function of pixel concentrations.

**Eq. 16**
* * *
65  *I still do not understand why Eq. 14 and the higher derivative was introduced. Eq. 16 describes a minimization problem and it should be fully sufficient to minimize \omega I(f). So why introduce Eq. 14? That the higher derivative is zero is a necessary condition for the existence of a minimum, but that is a detail of the employed minimizer. Unless a Newton-type-method is used, Eq.14 is not necessary and seems to complicate the computation. I assume that the algorithm will still work with this seemingly needless complication.*

70  Eq. 16 is the result of the derivation. If Eq. 14 is not introduced, then we will not get the matrix M in Eq. 16. It is true that we can also work on the original form of I(f) instead of $M$, and use a different optimization method to solve the minimization problem. In this study, Eq. 16 is actually the result after using the Newton-type-method. We use this form because it is similar with the formula of LTD algorithm. The only difference for the two approaches is the matrix $M$ comparing with $D_3$ which is used in the LTD approach. Thus, the same technique can be used to solve the minimization problem, which makes
75  the comparison more meaningful.

**Response to comments of reviewer 3**

We thank the reviewer for the helpful comments and suggestions, and for careful reading of the manuscript. Listed below are our itemized responses, with the original comment/question displayed in italics.

**General**

*As compared to the first version, the paper by Sheng Li, Ke Du has been significantly changed. Many of the concerns I pointed out in the first review have been addressed. I apologize that while reading the new version of the paper, unfortunately, I found some new relevant issues. Some of them are crucial and, to my opinion, should be carefully considered before publication.*

*In general the language has been greatly improved, however, I would recommend the authors to double check if the editing process has always conserved the original meaning of what they wanted to say.*

Thanks for the comments. We have revised the manuscript according to your comments. Please see the responses below for specific changes we have made to the manuscript.

**Specific comments**

**Sect. 2.1:**

*the section has been significantly improved. I see only two remaining issues:*

*a) Linearity of equation (2). Your statement at line 848 of your replies "The predicted PIC for one beam equals to the summation of the multiplication of the pixel concentration and the beam length passing it" holds only in the "optically thin" regime, i.e. only when a small fraction of the laser signal is absorbed by the pollutant to be measured. If the pollutant concentration is large and/or the medium is not very transparent at the laser wavelength, then for sure linearity does not hold. Since the laser wavelength can be tuned, usually it is possible to find a suitable wavelength at which, for average or expected pollutant concentrations, the medium is sufficiently transparent to fall in the linear regime. To my opinion it would be convenient to state explicitly that you are working in this hypothesis.*

Thanks for the suggestion. We are using open path TDL analyzer, which is already well-tuned equipment. The wavelength of the laser beam is tuned to the absorption line of the target gas and is transparent to other species. The laser absorption is in the linear regime and the attenuation of the laser beam is governed by the Beer–Lambert law. This introduction gives a description how the PIC is calculated by the equipment. So the condition of "optical thin" was satisfied in our field experiments. We also added a sentence on page 5 to state that our calculation was based on linearity between laser absorption and pollutant concentration.

*b) Equation (3): is it really needed to require c ≥0 ? Assume that the real concentration of the pollutant to be measured is zero. If you obtain the concentration as the average of several measurements, then, because of the analyzer's noise, these measurements should be evenly distributed around zero to give a zero average value. If you constrain the solution c to be greater or equal to zero, then you remove the measurements that otherwise would be negative because of the measurement noise. Thus you introduce a bias (or exposure error) in the average. Of course you don't see this type of error in the tests presented later in the paper, because the test 2D distributions considered do not include a "zero concentration" case, and because all your formulas fully ignore the measurement noise error (that is the noise error on your b vector).*

We understand that non-negative constraint may result in bias when averaging the concentrations where only noise exists. However, this is a trade-off as we need to focus on signals well above noise. Depending on the inverse problem, the optimal solution may contain very unrealistic negative value. This is especially true for the case in the 2-D tomographic mapping of air pollutants, whereas the data is limited, and the system of linear equations is ill-posed (e.g., rank-deficient). Therefore, it is necessary to apply these constraints to eliminate such unrealistic solutions because the resulted negative values are not just at noise level. They may be comparable with the positive peak values. In addition, there are different sources of errors. In the simulated study, one error source is discretization error. Even in this case, the results generated without non-negative constraints are not useable. After using the regularization technique, the optimized solution is robust to the measurement noise, and the reconstruction error is dominated by regularization error. Therefore, the measurement noise is negligible, and we can focus the study on the regularization algorithm itself. We have added a sentence below Eq. (3) to explain the necessity of using non-negative constraints.

**Sect. 2.2:**

*If I understand properly, in practice what you call LTD method is the Tikhonov regularization using the discrete 3rd derivative operator $D_3$. You tune the regularization strength w using the discrepancy principle. Then I have a question: you can not introduce the two equations (4) at every pixel of your domain, as you would have problems at the edges of the domain (see index values $k+2$, $k+1$ and $k-1$ in eq. 4). Thus, you can actually introduce only $2N_c - 6$ additional equations (not $2N_c$ as you state at line 160). If you do so I think you get in trouble if one or more pixels of your domain are not crossed by a ray path. Actually, in this case $L^tL$ would be singular and, if also $T^tT$ is singular, then you cannot invert the matrix $A^tWA$ mentioned at line 172. In conclusion, I think you are using some extra constraints (also involving 0-th order derivatives) at the edges of your domain (as you seem to suggest in the text after eq. 7). Please clarify. Also, what are the symbols m and n in eq. 7?*

***Line 243****: please explain clearly the equations you use at the edges of the domain, see also the analogous comment above.*

They were indicated at line 182 and 243 in the manuscript that the constraints at the edges are second-order and first-order difference operators. According to reviewer's comment, we have added their formulas to the manuscript. *m* and *n* are the row and column number of the pixels. They keep the same definition across the manuscript. It is fine if a pixel is not crossed by a ray path. With the high-resolution grid division, most of the pixels are not crossed by the beam. They are restricted by the smoothness constraints.

*Line 245: requiring $c \geq 0$ introduces a bias (i.e. a systematic exposure error) for very small values of the real concentration distributions c, see also the analogous comment above.*

The non-negative constraints are necessary to eliminate the unrealistic negative solutions resulting from measurement errors and ill-posed inverse problem. Please see the response in Sect. 2.1. (b).

**Sect. 2.4:**

*Here you define the "true" concentration distributions that will be considered later to assess the various reconstruction methods. You should say also something regarding the synthetic observations b that you build on the basis of g(x,y). Which is the spatial mesh (or grid) that you use for the calculation of b, starting from g(x,y)? Do you add pseudo-random measurement noise to b, to emulate the real world situation in which the instrument is subject to measurement noise? Your equations discard the measurement error. Mathematically this is equivalent to assume uncorrelated noise equal to 1, thus you should add to each element of b a pseudo random noise with zero average and standard deviation equal to 1.*

The concentration filed is discretized with a resolution of 0.2 m×0.2 m. The concentration of each pixel is the average value of the concentrations in that pixel. The discretized concentration map is used as the true concentration distribution. PICs are calculated based on the discretized map by using Eq. (1). We have added this description to the Sec. 2.4.

160     The purpose of this study was to study the different regularization techniques to produce smooth reconstruction, instead of studying the influence of the measurement noise. After using the regularization technique, the optimized solution is robust to the measurement noise, and the reconstruction error is dominated by regularization error. Therefore, the measurement noise is negligible.

165     ***Lines 293 and ff****: the unusual naming conventions used in this part make it very difficult to understand for me and, I guess, also for the whole atmospheric community. G matrices defined at lines 298-300 are usually called "gain matrices". It is true that in this case they also represent the "generalized inverse" of L because the forward model (Le) is linear. Despite of that, I would still call G the "gain matrix", just for uniformity with Rodgers 2000. Equations at lines 299 and 300 should contain a"+" sign within the parenthesis. Matrix R is usually called "averaging kernel", not "resolution matrix", just because it*
170 *shows how a change in the real concentration field maps onto the concentration estimated by the inversion system. The "resolution" is usually a scalar or vector quantifier computed on the basis of R. Line 304: what you call "regularization error" is usually called "smoothing error".*

The names the reviewer gives are common in the atmospheric sounding problem. But the notations we use are more common in the problem of 2-D or 3-D tomography. It may be necessary to discuss the differences between the atmospheric sounding
175     problem and 2-D gas tomography.

1)     One of the differences is apparently the 1-D inversion versus 2-D inversion. In the 2-D tomographic applications, a new factor to consider is the geometric configuration of the beam paths, whereas one does not need to do in the 1-D case. As a result, the path geometry largely affects the inversion. And this influence can be seen from the averaging kernel matrix (see Fig. 4 in the manuscript).

180 2)     For retrieval of vertical profile of an atmospheric quantity, a priori profile ($x_a$) of the unknowns is usually introduced. The error caused by the priori profile can be measured by using the averaging kernel matrix, which makes it an important tool to evaluate the sensitivity of the inversion. However, in 2-D gas mapping problem, there is no *a priori* distribution of the gas concentration introduced. Therefore, the significance of the averaging kernel matrix is reduced.

3)     The averaging kernel matrix is also used to evaluate the regularization error, which is determined by the inversion
185     algorithm instead of the a priori profile. This is what we did in this study. In this case, the regularization term forces the off-diagonal terms to be nonzero, thereby making the estimated concentration of each pixel a weighted average of the concentration of the surrounding pixels. As a result, there is no need to pursue a prefect averaging kernel matrix. The smooth weighting of the matrix is intended and necessary. This is also the focus of this manuscript.

In summary, we think there are differences between the two applications fields. We think it is a hard choice and decide to
190     change the 'resolution matrix' to 'averaging kernel matrix' and keep other names unchanged.

***Line 305****: what is the perturbation error G delta_b? Why should the observed quantities be perturbed? Maybe delta_b is the measurement error (noise + calibration, etc.)? Please explain.*

The perturbation is the measurement error due to various noise sources. We have updated the description in the manuscript.

195

***Line 307****: note that the non - sensitivity to the "perturbation error" is not always an appreciated property for the solution. In fact this means that the solution is mostly independent from the observed quantity, therefore it mostly depends on the applied constraints, that is exactly what one usually wants to avoid.*

As indicated in the manuscript (section 2.5) and in the response in Sect. 2.1. (b), the study focuses on the regularization term, which dominates the reconstruction error comparing to the perturbation error. In this case, the regularization error cannot be avoided and can be used to evaluate different regularization techniques. Please refer to the discussion of the differences between the atmospheric sounding problem and the 2-D gas mapping problem (Sect. 2.4. Lines 293 and ff).

*Eq. 23: why don't you write explicitly the expression for the Frobenius norm? Which is the advantage of using the Frobenius norm over the more usual trace[R] that, according to Rodgers, 2000 would represent the number of degrees of freedom for the signal? If R is not a orthogonal matrix (like in this case, I guess) then I am not able to find an easy interpretation for your quantifier of eq.(23)*

Frobenius norm is the square root of the sum of the absolute squares of the elements of a matrix. We have added the definition to the manuscript. $R$ is not an orthogonal matrix and its row contains the weights of all pixels, which are determined by the path geometry and the regularization algorithm discussed in the response of Lines 293 in Sect. 2.4. As a result, the weights are spread over the pixels, and trace[R] is not useful in the study of either the regularization error or the path geometry, which is why we use the Forbenius norm.

*Line 365 and ff: how do you interpret the systematically slightly worse performance of MC as compared to LTD in terms of exposure error? I am also surprised to see such a large exposure error in the NNLS method as compared to the other two methods. Maybe this due to the larger size of the pixels used with NNLS? In this case it would be better to change eq.(20) by multiplying each concentration by the area of the pixel to which it refers (actually eq.(20) should compare the integrals of the concentration in the considered domain).*

After carefully examining the results. We found that that large error of NNLS and slightly worse performance of MC were mainly caused by using the spline interpolation after the reconstruction. Therefore, we have reproduced the results by using the nearest which has minimal effect on the original results. The new results illustrate that the MC algorithm shows slightly better performance than LTD in all cases, and the performance of NNLS is greatly improved than the previous results. We have updated Table 3 and the discuss of the results.

Eq. (20) already includes the integrals of all the pixels in the domain. All the reconstructed pixels are compared with the simulated 'true' pixels. Therefore, they are using the same number of pixels no matter what their original grid divisions are.

**Sect. 3.4:**

*at line 144 of the revised paper finally you reveal that you are using the "lsqnonneg" MATLAB routine to minimize the cost functions relating to the various methods considered. I was not able to establish what is exactly the minimization method used by this MATLAB function, however, for sure it does not find the solution mentioned at line 172 of your revised manuscript, because this latter is not bound to be positive. As explained later, I suspect that lsqnonneg is using iterations, which is not optimal at all for the linear case (b = Le) you are dealing with. This implies that the run-time analysis presented in sect. 3.4 is applicable only to the unlucky case in which one uses lsqnonneg. The optimal solution of your inversion would be to implement, directly, in a computer program the matrix operations at line 172. This would make the computation time dependent only on Nb (number of measured PICs) and on Ne (number of unknowns). Both Nb and Ne do not depend on the number of sources that you put in your domain, thus the computation time would not depend neither on the number of sources considered nor on their amplitude and location in the domain. You also claim that the MC method is much faster with respect to the LTD because of the fewer constraint equations. To my experience, if the solution at line 172 would be implemented, most of the computation time would be spent in the inversion of matrix A1WA, of dimension Ne. The number of constraint equations used would impact only the calculation of the product TT (that is needed to compute A'WA), thus the total computation time should depend only very marginally on the number of constraints used.*

The NNLS algorithm of 'lsqnonneg' uses the algorithm described by Lawson and Janson (1995). As indicated in the manuscript, it is an active-set optimization method using an iterative procedure to converge on the best fit of positive values. The analytical solution is only demonstrated for the case without non-negative constraints. But it is not applicable in the problems with non-negative constrains and was not used in this study. Because of this, an iterative optimal algorithm needs to be used to solve the constrained inversion problem. Thus, your concern regarding computation time of *lsqnonneg* routine is not applicable in this case. We have added a sentence in the paragraph under Eq. (5) to explain this.

*Another discussion point are the absolute values of the computing times shown in Table 4 and their standard deviations: why the computation time changes and thus shows a standard deviation also when the method used and the number of sources do not change? (I suspect this is because lsqnonneg uses iterations whose number changes from test to test). Why is computation time so long? Are you using a very old CPU? Please specify. Assuming your inversion problem dimensions, finding the solution given at line 172 would require less than 1 second with a standard CPU. In conclusion, finding the solution as c = (AT W AJ1 AT W p ( as specified at line 172) would be much faster and the savings in computing time achieved with the MC method over the LTD would be much less important*

lsqnonneg uses an iterative procedure to converge on the best fit of positive values. Therefore, its speed is affected by the values of the coefficient matrix. This is common for an iterative optimal algorithm. We used a modern computer to do the calculation. The configuration of the computer has been listed according to the comment. The fast speed the reviewer gives is based on the analytical formula, which is not applicable in this study as explained in previous response.

**Line 378***: what do you infer from fig's 4a and 4b? Are they useful? I was not able to extract information from them. In my previous review I was suggesting to include only Fig.s 4e and 4f. Of course additional figures are welcome, however they should convey useful information that should also be discussed in the text.*

The first two figures are the visualization of the fitness function. It is in 2-D form. Therefore, the values are represented by colors, and we cannot expect a curve like that in the 1-D inversion. We have revised the description.

**Line 388***: figures 4e and 4f show diagonal elements significantly smaller than 1, this means that your regularization is actually very strong (w is very large). What do you get for trace[R] / Ne? I guess it is < < 1. In practice I feel that with a much softer regularization you would get solutions with better accuracy and similar smoothness.*

As indicated in the responses of Eq. 23 in Sect. 2.4, the elements in a row of the averaging kernel matrix are determined by the path geometry and regularization terms. The purpose is to evaluate the regularizations and this observation implicates it is working. The trace is not helpful in this case and fitness value was used.

*Figure 4: at least, please use the same color scale in the left and right maps, otherwise the comparison of the two methods is very hard ....*

The figures have been updated according to the suggestion.

**Section 3.6:**

*the results reported in this section are not useful as they are not general for the experimental setup considered, they depend on the dimension of the assumed sources. From what stated at line 399, I understand that these sensitivity tests are performed assuming 5 Gaussian source functions, whose minimal Full Width at Half Maximum (FWHM) is 2.8 m x 2.355 = 6.59 m (see line 272 and the properties of the Gaussian function). When the dimension of the smallest source function is about 6.6 m, clearly there is no much advantage to go from a 18x18 grid (pixel size = 40m/18 = 2.22 m) to a 40 x 40 grid*

285 *(pixel size = 40m/30 = 1.33 m) as in both cases the pixel size is significantly smaller than the source size. Usually, in real situations, the size of the source is unknown and what is useful to know is the spatial resolution of the measuring plus inversion system. The averaging kernels are usually an adequate tool to answer this question: the dimension of the main yellow spots of maps in fig. 4c and 4d quantify the spatial resolution of your measuring system at the location of indexes 16 and 4.*

290 We used the same settings of the sources with the GT-MG, whereas the values are common for mapping of small-scale air pollutants. These results generally show us the trends with changing grid size. It was also indicated in the manuscript that there is a certain threshold value from the trend, which was 24×24 in this case. It may be a different resolution for other application, but the trends are the same. Fig 4. (c) and (d) clearly show the averaging kernels mainly determined by the beam geometry, which makes it a good tool for optimizing the beam configuration.

295 *Lines 410 - 413: from the theoretical point of view, the growth of computation time for this inverse problem should roughly follow a polynomial law as a function of the number of pixels. Did you really prove that the behavior you find is exponential? If the behavior is really exponential then you should give a justification so as why the computation time increases so rapidly in your case.*

The exponential trend was concluded from an exponential fit with a $R^2=0.9996$. Our purpose is to show that the computation
300 time grows fast with increasing pixels and the trend is 'approximately' exponential. Of course, we can fit it with a polynomial (dependent on what order you use). But we are not intended to prove what exactly the trend it should follow, which is related to the details of the optimization algorithm. What we did was to use the same optimization algorithm for all the three reconstruction techniques presented in this study to make sure the comparisons were based on the same standard.

305 *Conclusion section: as outlined above, in my view the time computing savings of the MC method, if any, should be assessed using an optimal solving algorithm for your inversion problem, not the MATLAB lsqnonneg routine. In conclusion, either you renounce claiming the time savings or you implement an optimized solver (like eq. at line 172 implemented in FORTRAN or in C).*

As explained in the response of the second question in Sect. 3.4 responses, the analytical formula is for the problems without
310 non-negative constraints and is not applicable to the problems with non-negative constraints in this study. Also, the NNLS algorithm that the lsqnonneg uses is the most widely used inversion algorithm for solving non-negative least squares problems.

**Minor corrections**

315 **Line 14:**

*I would say "regularization term" (regularization is not a factor in the inversion formula)*

"term' has been used.

**Line 22 (and line 113):**

320 *It is also simpler to perform .. . maybe you meant "implement" instead of "perform" ?*

'implement' has been used.

**Line 204:**

*bending ?*

325   Changed to "potential energy stored in a bended elastic object."

**Line 272:**

*units of sigma_x and sigma_y should be "m" (meters).*

The unit 'm' has been added.

330   **Line 416:**

seminors ? ·

'seminorms' has been used.